

# Identification of Parameter Importance for Benzene Transport in the Unsaturated Zone Using Global Sensitivity Analysis

Meirav Cohen[1], Nimrod Schwartz[2], Ravid Rosenzweig[3]

[1] The Dead Sea & Arava Science Center (DSASC), Mitzpe Ramon, Israel
10 [2] The Institute of Environmental Sciences, The Robert H. Smith Faculty of Agriculture, Food and Environment, The Hebrew University of Jerusalem, Rehovot 76100 , Israel
[3] Geological Survey of Israel, Jerusalem, Israel[2]

15  *Correspondence to*: Meirav Cohen (meirav@adssc.org)





**Abstract.**

One of the greatest threats to groundwater is contamination from fuel derivatives. Benzene, a highly mobile and toxic fuel derivative, can easily reach groundwater from fuel sources and lead to extensive groundwater contamination and drinking water disqualification. Modelling benzene transport in the unsaturated zone can provide quantification of the risk for groundwater contamination and needed remediation. Yet, characterization of the problem is often complicated, due to typical soil heterogeneity and numerous unknown site and solute parameters, as well as the difficulty in distinguishing important from non-important parameters, resulting in high uncertainty. Thus, the application of sensitivity analysis (SA) methods, such as global SA (GSA), is required to reduce uncertainty and detect important parameters for groundwater contamination, mitigation, and remediation. Nevertheless, studies devoted to identification of the driving parameters for fuel derivatives transport in the unsaturated zone are scarce. Here, we performed GSA on a problem of benzene transport in the unsaturated zone. First, a simple GSA 'Morris' screening method was used for a homogenous sandy vadose zone. Then, a more computationally-demanding 'Sobol' variance-based GSA was run on the most influential parameters. Finally, the 'Morris' method was tested for a heterogeneous medium containing clay layers. To overcome the problem of model crashes during GSA, several methods were tested for imputation of missing data. The GSA results found benzene degradation rate ($\lambda_k$) to be the utmost influential parameter controlling benzene mobility. The depth of aquifer followed in importance in the homogenous media, while in the heterogeneous media parameters related to the clay layers, such as clay adsorption coefficient and the number of clay layers, followed in importance. The study emphasizes the significance of $\lambda_k$ and the presence of clay layers in predicting aquifer contamination. The study also strengthen the importance of heterogenous media representation in the GSA, since different parameters control the transport in different soil layers. Overall, GSA is demonstrated here as an important tool for the analysis of transport models. The results also show that in higher dimensionality models, the radial basis function (RBF) is an efficient surrogate model for missing data imputation.





## 1. Introduction

Petroleum products such as gasoline and diesel are of the most abundant chemicals of ecological concern used nowadays. During petroleum exploration, production, transport and storage, petroleum products often find their way to the environment by accidental leaks and spills (Logeshwaran et al., 2018; López et al., 2008; Nadim et

al., 2000). Consequently, groundwater is often polluted from surface sources, posing a substantial potential risk to potable water worldwide (López et al., 2008; Nadim et al., 2000; Logeshwaran et al., 2018; Reshef and Gal, 2017; Kessler, 2022). Since petroleum substances in general, and fuel components in particular, are considered toxic, carcinogenic, and mutagenic (Logeshwaran et al., 2018), strict regulations limit their maximum allowed concentration in groundwater to the parts per billion level (U.S EPA, 2006).

Fuel products are usually comprised of different types of hydrocarbons. Benzene specifically, is highly soluble and thus of the most mobile fuel constituents in the subsurface (Farhadian et al., 2008). In Israel, for example, benzene was detected in 60% of all sites monitored for fuel contamination (Reshef and Gal, 2017). Benzene low maximum allowed concentration of 5 μg/l in drinking water, arises great concerns that benzene leakages into groundwater will lead to disqualification of extremely large volumes of drinking water (Kessler, 2022). Most

fuel-related contamination reaches groundwater from the soil surface or near it (Troldborg et al., 2009). Therefore, it is important to assess the risk to groundwater from soil contamination and to understand the fate and transport of fuel components travelling from the soil surface, through the unsaturated zone, and down to the groundwater.

Most models simulating the transport of fuel contaminants in the unsaturated zone use a mechanistic description

of the physical, chemical, and biological processes controlling contaminants' transport and attenuation. These models include many uncertain input parameters, due to the typical heterogeneity of the subsurface environment and the difficulty in obtaining sufficient relevant physical and bio-geochemical characterization of the site (Tartakovsky, 2007). Thus, sensitivity analysis (SA) is required to determine the contribution of the individual



input parameter to the uncertainty of the model output (Song et al., 2015). More specifically, SA can determine

which are the non-influential input parameters that can be redundant or fixed, reveal the order of parameter

importance, and the magnitude of parameter interactions (Razavi et al., 2021).

Various SA methods are available; they can be generally divided into local sensitivity analysis (LSA) and global

sensitivity analysis (GSA). LSA methods are mostly "one at a time" (OAT) methods. These methods are based

on changing the uncertain input parameter by a specific interval several times around a 'local point' in the

problem space. The difference or the derivative of the output compared with the base-case output is then tested.

LSA methods are usually simple and efficient in analyzing simple models, but they are less suitable for multiple

parameters, non-linear and non-additive models. This is because derivatives are informative at the base point

where they are computed, but do not enable exploration of the rest of the input parameter space (Saltelli and

Annoni, 2010). Moreover, OAT methods are efficient in finding the most influential input parameters but cannot

rank the influence of the input parameters or measure parameter interactions (Saltelli and Annoni, 2010). In

GSA, on the other hand, the input parameters are changed over the entire sampling space, and the variance of

the output is tested rather than the derivative. In addition to finding the most influential input parameters, GSA

can rank the parameters' influence and their interactions (Saltelli and Annoni, 2010; Razavi et al., 2021). Yet,

the main drawback of GSA methods is their computational cost.

Most hydrological models have numerous parameters resulting in high-dimensional and nonlinear problems.

Therefore GSA methods are usually recommended in hydrological modelling (Song et al., 2015). Song et al.

(2015) recommends GSA application before final modelling in order to better understand the model and its

dominating parameters, and as a tool to reduce the model parametric dimensionality. So far, application of

LSA/GSA for contaminant transport in the unsaturated zone is scarce (Davis et al., 1994; Gatel et al., 2019;

Gribb et al., 2002; Pan et al., 2011; X Song & Chen, 2018; Ciriello et al., 2013, Ciriello et al., 2017), where

most papers considered a homogenous media. Specifically for benzene, few LSA analyses indicated the



degradation and adsorption coefficients as the most important parameters for benzene transport in the unsaturated zone (Gribb et al., 2002; Zanello et al., 2021). In one GSA performed for benzene risk assesment to groundwater (Ciriello et al., 2017), the porosity and hydraulic conductivity of the media were found to be the

dominant parameters for the model uncertainty. Yet, the proporties of benzene itself were treated as deterministic quantities, and were not tested by the GSA (Ciriello et al., 2017). Moreover, we are not aware of any SA performed for benzene transport in layered heterogenoues unsturated media. Hence, there is a need for more research on the key parameters controlling contaminants transport in the subsurface, in general, and for benzene, in particular, for better prediction and mitigation of groundwater contamination.

The objective of this study was to assess the individual impact of multiple parameters on benzene transport in the unsaturated zone. For that purpose, a mechanistic model was used to simulate the transport of benzene in an unsaturated zone representing Israel's coastal plain vadose zone. Simulations of both homogenous (sand) and heterogeneous (sand with clay layers) vadose zones were conducted. Two GSA methods were tested for the homogenous media simulations, to analyze the parameter importance: the Morris method (Morris, 1991), a

reliable, computationally-cheap alternative to variance-based GSA; and the Sobol method (Sobol, 2001), a computationally heavy, variance-based GSA. The heterogeneous media simulation was tested by the Morris method, where the effect of the parameters of both soil types was tested as well as the clay layers' distribution.

A common problem in GSA application is that some of the model runs do not converge but crash due to numerical instability and the assignment of random sets of parameters of different values. Owing to the novelty of GSA in hydrological research, there is not one agreed and established way to deal with these missing data,

of GSA in hydrological research, there is not one agreed and established way to deal with these missing data, and the information in the literature is still scarce (Sheikholeslami et al., 2019). Consequently, and as part of the overall analysis done in this study, we also tested several methods for missing data imputation in cases where the model does not converge or crashes.



## 2. Materials and methods

A mechanistic model was generated to investigate the potential transport of benzene in the vadose zone underlain by Israel's coastal plain aquifer. The model was applied for both homogenous and heterogeneous media. For simplicity, initial runs solely included a homogenous sandy soil profile, as sand and sandstone are the main constituents of Israel's coastal plain aquifer (Kurkar Group; Turkeltauub, 2011). Yet, in most natural environments, the soil profile is non-homogenous, containing clay layers and other materials. In a study

conducted at Israel coastal plain vadoze zone, Rimon et al. (2007) reported that the occurrence of different soil materials, and specifically clay interbeds, strongly affected flow patterns to the aquifer. Therefore, later model runs included clay interbeds.

### 2. 1 Mechanistic model and input parameters

The one-dimensional mechanistic model included water flow, solute transport, biodegradation, adsorption, and

volatilization. Water flow in the unsaturated zone was modelled by a modified form of Richard's one - dimensional equation,

$$\frac{\partial \theta}{\partial t} = \frac{\partial}{\partial z}\left[K\left(\frac{\partial h}{\partial z} + 1\right)\right] \tag{1}$$

where $h$ is the water matric head [L], $\theta$ is the volumetric water content, $t$ is time [T], $z$ is the vertical coordinate [L] (positive upward), and $K$ is the unsaturated hydraulic conductivity function [LT$^{-1}$] given by,

$$K(h,z) = K_s(z)K_r(h,z) \quad\quad 125 \tag{2}$$

where $K_r$ is the relative hydraulic conductivity [-] and $K_s$ the saturated hydraulic conductivity [LT$^{-1}$].

The unsaturated soil hydraulic properties $\theta(h)$ and $K(h)$ are described by the van Genuchten – Mualem formulation (Mualem, 1976; van Genuchten, 1980),

$$\theta(h) = \theta_r + \frac{\theta_s - \theta_r}{[1 + |\alpha h|^n]^m} \quad\quad h < 0 \tag{3}$$

$$K(h) = K_s S_e^l \left[1 - \left(1 - S_e^{1/m}\right)^m\right]^2 \quad\quad 130 \tag{4}$$



$$S_e = \frac{\theta - \theta_r}{\theta_s - \theta_r} \tag{5}$$

where

$$m = 1 - 1/n , \ n > 1 \tag{6}$$

In Eq. (3-6), $\theta_s$ is the saturated water content and $\theta_r$ is the residual water content. $\alpha$ [L$^{-1}$], $n$ and $m$ are the van

Genuchten fitting parameters; $S_e$ is the effective saturation, and $l$ is the pore-connectivity parameter (Eq. (4)).

Solute transport was described by the advection-dispersion equation,

$$\frac{\partial \theta c}{\partial t} + \rho \frac{\partial s}{\partial t} + \frac{\partial a_v g}{\partial t} = \frac{\partial}{\partial z}\left(\theta D^w \frac{\partial c}{\partial z}\right) + \frac{\partial}{\partial z}\left(a_v \tau_g D^g \frac{\partial g}{\partial z}\right) - \frac{\partial qc}{\partial z} - \lambda_k \theta \tag{7}$$

where $c$, $s$, and $g$ are solute concentrations in the liquid [ML$^{-3}$], solid [MM$^{-1}$], and gaseous [ML$^{-3}$] phases,

respectively. $\rho$ [ML$^{-3}$] is the solid phase bulk density. $D^w$ is the dispersion coefficient in the liquid phase [L$^2$T$^{-1}$]

given by Bear (1972) as,

$$\theta D^w = \alpha_L q + \theta D^M \tau_w \tag{8}$$

where $D^M$ is the benzene's molecular diffusion coefficient in the aqueous phase [L$^2$T$^{-1}$], $q$ is the absolute value

of the Darcian fluid flux [LT$^{-1}$] evaluated using the Darcy–Buckingham law, $q = -K(\frac{dh}{dz} + 1)$. $\alpha_L$ is the

longitudinal dispersivity [L], and $\tau_w$ and $\tau_g$ are tortuosity factors in the liquid and gas phase respectively [-],

evaluated using the relationship described by Millington & Quirk (1961). $a_v$ is the air content [L$^3$L$^{-3}$], and $D^g$ is

the benzene molecular diffusion coefficient [L$^2$T$^{-1}$] in the gas phase and $\lambda_k$ is a first-order rate biodegradation

constant for benzene in the liquid phase [T$^{-1}$] (solid and gas phase degradation were assumed negligible).

Benzene adsorption was assumed linear (Wołowiec and Malina, 2015; Baek et al., 2003) of the form $s = K_d c$,

where $K_d$ is the distribution coefficient [L$^3$M$^{-1}$] (see Table S2 in the appendix for literature values).

The gaseous (g) and aqueous (c) phase concentrations in Eq. (7) are related by a linear expression of the form:

$$g = k_g c \tag{9}$$


where $k_g$ is an empirical constant [-] equal to $(K_H R_u T^A)^{-1}$ (Stumm and Morgan, 1981), in which $K_H$ is Henry's law constant [MT$^2$M$^{-1}$L$^{-2}$], $R_u$ is the universal gas constant [ML$^2$T$^{-2}$K$^{-1}$ M$^{-1}$] and $T^A$ is the absolute temperature [K].

The values of $\theta_s, \theta_r, l, D^M, D^g, K_H$, and $\rho$ were kept constant in the model and are listed in Table 1. $D^M$, $D^g$ and $K_H$ are constant properties for benzene and were therefore not changed. The pore-connectivity parameter $l$ in the hydraulic conductivity function was estimated  to be about 0.5 as an average for many types of soils (Mualem, 1976). The range of $\theta_s$, $\theta_r$, and $\rho$ values in the literature was limited (Carsel & Parrish, 1988; Domenico & Schwartz, 1990; Gribb et al., 2002; Schaap et al., 2001). In a local sensitivity analysis of benzene

transport in the vadose zone, Gribb et al. (2002) found that $\rho$ is an insignificant parameter and $\theta_r$ is only significant in pure clayey soils.

The sensitivity of the model to the values of $\alpha, n, K_s, \alpha_L, \lambda_k$ and $K_d$ values was tested in the GSA analysis. The range of tested values along with the corresponding references can be found in Table 2. Specifically, we found that $\lambda_k$ values greatly vary between different studies, mainly due to the differences in experimental conditions

and aquifer characteristics (See appendix Table S1 for literature values). Though the highest $\lambda_k$ value we encountered in the litreture was 174 (day$^{-1}$) (Lahvis et al., (1999); Table S1), we set the upper limit of $\lambda_k$ to 1.5 (day$^{-1}$) (Table 2). This was done for two main reasons; A. from an early stage it was evident that $\lambda_k$ is a very influential parameter and high values mostly resulted in output values of zero, thereby lowering the overall sensitivity. B. The Morris analysis takes the range and divides it into a given number of levels (four or six, in

our case). Since the range of $\lambda_k$ included values spanning over four orders of magnitude ($1 \times 10^{-2} - 1 \times 10^2$ (day$^{-1}$)), much of the range would have been missed by the analysis.






**Table 1** Constant input parameters for the model:

| Parameter | value | units | Reference |
|---|---|---|---|
| $l$ | 0.5 | -- | Mualem (1976) |
| $\theta_r$ Sand | 0.045 | -- | Carsel & Parrish (1988) |
| $\theta_s$ Sand | 0.43 | -- | Carsel & Parrish (1988) |
| $\theta_r$ Clay | 0.068 | -- | Carsel & Parrish (1988) |
| $\theta_s$ Clay | 0.38 | -- | Carsel & Parrish (1988) |
| $\rho$ | 1500 | kg/m$^3$ | Levy (2015) |
| $D^m$ Benzene | 7.77E-05 | m$^2$/day | EPA On-line Tools for Site Assessment Calculation |
| $D^g$ Benzene | 0.77414 | m$^2$/day | EPA On-line Tools for Site Assessment Calculation |
| $K_H$ Benzene Henry's constant | 0.224 | -- | Du et al. (2010) |



**Table 2** Input parameters and their range used for GSA:

| | Sandy soil | | | Clayey soil | | |
|---|---|---|---|---|---|---|
| Parameter | Values range | References | Comments | Values range | References | Comments |
| $\alpha$ (m$^{-1}$) | 3.6 - 17 | Carsel & Parrish, (1988), Domenico & Schwartz. (1990), Moret-Fernández et al., (2017), Nemes et al., (1999) | Fine to coarse sand Silt to sand | 0.5 - 3.86 | Carsel & Parrish. (1988) Rawls et al. (1982) | Clay to clay loam |
| $n$ | 1.1 - 2.9 | Domenico & Schwartz, (1990); Moret-Fernández et al. (2017); Nemes et al. (1999) | Fine to coarse sand | 0.13 - 1.31 | Carsel & Parrish, (1988); Rawls et al. (1982) | Clay to clay loam |
| $K_s$ (m/day) | 0.5 - 250 | Domenico & Schwartz, (1990) Nemes et al., (1999) | Values for fine to coarse sand are 0.017-518 (m/day). These were narrowed for model convergence (coastal aquifer typical values are ~7 m/day) | 0.001 - 0.3144 | Rawls et al. (1982). | Clay to clay loam |





| | | | | |
|---|---|---|---|---|
| $\alpha_l$ (m) | 0.01 - 5 | According to a rule of thumb 1/10 of z (Bear & Cheng, 2010) | 0.01 - 5 | According to a rule of thumb 1/10 of z (Bear & Cheng, 2010) |
| $K_d$ (m³/kg) | 0.00007 - 0.004 | Literature review for benzene in sandy soils (Table S2) | 0.00004 - 0.0238 | Literature review for benzene in clayey soils (Table S2) |
| $\lambda_k$ (day⁻¹) | 0 - 1.5 | Literature review for benzene in unsaturated media (Table S1). Upper value was set to 1.5 to obtain output results >0 | 0 - 1.5 | Literature review for benzene in unsaturated media (Table S1). Upper value was set to 1.5 to obtain output results >0 |
| $N$ | - | - | - | 1 - 4 | See Table 7 |
| $b$ (m) | - | - | - | 0.2 - 2 | See Table 7 |
| $z$ (m) | 5 - 50 | Typical aquifer depths | 10 - 50 | Lower value was increased for to allow space for clay layers insertion |





### 2. 2 Model domain and boundary conditions

The profile depth ($z$) was set as a variable input parameter in the range of 5 - 50 m (Table 2). This range was obtained from a

dataset of fuel-contaminated sites of the Israeli coastal plain received from Israel's Ministry of Environmental Protection and

reported by Israel's Water Authority (Reshef and Gal, 2017). In runs that tested the occurrence of clay layers, the thickness

($b$) and number ($N$) of clay layers were additionally tested as variable GSA input parameters (Table 2).

An upper atmospheric boundary condition (BC) was set at the top of the profile with average daily precipitation and

potential evaporation data from the Beit Dagan meteorological station for 2019 (Fig. 1). Potential pan evaporation data was

converted to Penman-Montieth potential evaporation by multiplying the data with monthly coefficients obtained for the

Israeli coastal plain by Gal et al. (2012). On days when evaporation data were not available, a monthly-averaged evaporation

value of the available data for the specific month was used as input. At the bottom boundary, where the aquifer was

positioned, a Dirichlet BC of constant matric head ($h = 0$) was set.

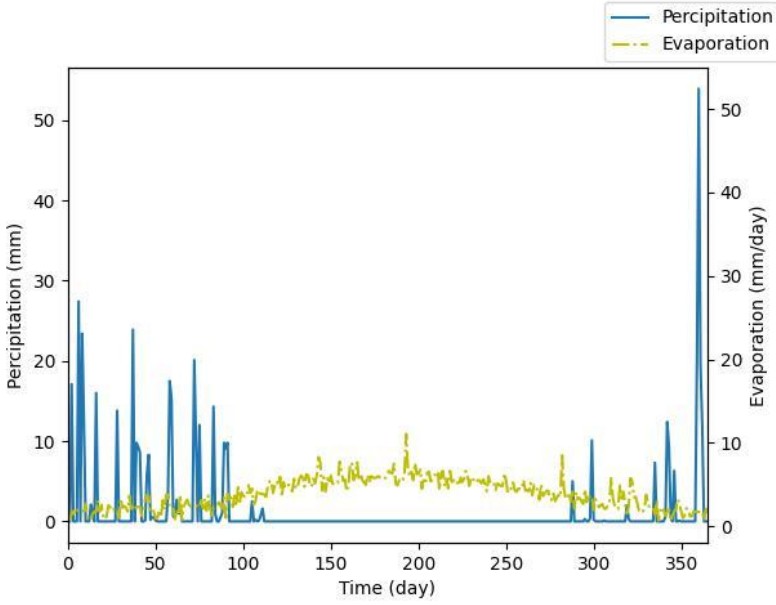


**Figure 1 – Daily precipitation and potential evaporation data of Beit Dagan meteorological station for 2019 – set as the upper model BC**

For the first ten years, only water flow was considered, to enable stabilization of the hydraulic conditions in the profile and establish annual periodic conditions. According to our tests, such stabilization takes about four years. Hence, benzene was introduced following ten years.

For the solute transport, an upper Dirichlet BC prescribing benzene saturation concentration (1.77 kg/m$^3$ - solibiluty of benzene in water at 25$^0$ C; Stewart. (2010)) was set to mimic a constant fuel lens on the surface. A bottom Neumann BC of zero concentration gradient was set, enabling free drainage to the aquifer.

The model was run using the Hydrus - 1D software package (Šimůnek et al., 2013), a finite element model for simulating the one-dimensional movement of water and solutes in variably saturated media.

In the homogenous media analysis, the soil profile was divided into 51 equal nodes. Yet for the heterogeneous media where layers of clay were introduced, a higher resolution was required due to the heterogeneity of the profile and the smaller size of the layers, as compared to the whole profile. In these runs, the profile was divided such that the total number of nodes was equal to $((z * 20) + 1)$. Clay layers were assigned in the profile according to the number of clay layers ($N$) and their thickness ($b$), such that they were equally distributed in the profile, generating alternating sand and clay layers. Each of the layers of both clay and sand were divided into $(b * 20)$ nodes.

The model was run for a total of either 100 years for the Morris analysis, or 60 years for the Sobol analysis (in this case, the number of years was reduced to save computational cost due to the overall high computational cost of the Sobol analysis). At the end of each run, benzene concentration in the aquifer and the total flux to the aquifer were examined.

### 2. 3 Global sensitivity analysis

Overall, three global sensitivity analyses tests were run on the model: two Morris analyses, one for homogenous and another for heterogeneous media, and one Sobol analysis for the homogeneous media.



### 2.3. 1 The Morris method

The Morris or the elementary effect (EE) method was introduced by Morris (1991) and improved by Campolongo et al.
(2007). It can be viewed as an extension of the OAT method, since it randomly generates sets of reference values from the

entire parameter space and computes the difference of output (EE) caused by a fixed parameter change, altering only one

parameter at a time. However, it can also be viewed as a GSA method, since it averages multiple EEs computed at different

points in the parameter space. This method provides qualitative sensitivity measures (i.e., ranking the input parameters in

order of importance), however it does not quantify how much a given parameter is more important than another (Saltelli et
al., 2004).

In the Morris method, each input parameter $x_i$, where $i = 1, \ldots, k$, is assumed to vary across $p$ selected levels in the space of

the input parameter. The parameter space is normalized to a uniform distribution in $[0, 1]$ and partitioned into $(p - 1)$

equal sections. The algorithm starts at a randomly chosen point in the $k$-dimensional space and creates a trajectory (or a

path) through the $k$-dimensional variable space. Each parameter is randomly chosen from the set $(p - 1)$ sections and a
fixed increment $\Delta$ (a multiple of $1 / (p - 1)$) is added to each parameter in random order to compute an EE of each

parameter, where EE is the difference of output $y$ caused by the change $\Delta$ in the respective parameter. The EE for the $i^{\text{th}}$

input parameter can be described as,

$$EE_i(x_1 \ldots x_k) = \left( \frac{Y(x_1, x_2, \ldots, x_i + \Delta_i, \ldots, x_k) - Y(x_1, x_2, \ldots, x_i, \ldots, x_k)}{\Delta_i} \right) \tag{10}$$

Changing each parameter once from one set of reference values completes one path, which together with the base case
requires $(k + 1)$ simulations. By conducting simulations over multiple paths, we have an ensemble of $EE_i$s for each

parameter. The number of required runs is then $r(k + 1)$, where $r$ is the number of paths or trajectories. All $EE_i$ values

computed for randomly chosen paths are used to compute two final sensitivity measures $\mu_i^*$ and $\sigma_i$ (Campolongo et al.,

2007),

$$\mu_i^* = \frac{1}{r} \sum_{j=1}^{r} |EE_{i,j}| \tag{11}$$



where $\mu_i^*$ is the mean of absolute values of the $EE_i$. $\mu_i^*$ can be regarded as a global sensitivity index, since it represents the average effect of each parameter over the parameter space. Thus, it is used to identify influential and non-influential parameters.

The second measure $\sigma_i$ is the standard deviation of the $EE_i$,

$$\sigma_i = \sqrt{\frac{1}{r}\sum_{j=1}^{r}\left(EE_{j,i} - \frac{1}{r}\sum_{j=1}^{r}|EE_{j,i}|\right)^2} \tag{12}$$

It is used to identify non-linear and/or interaction effects.

The review by Song et al. (2015) reported that in different studies, the number of paths ($r$) varies from 20 to 1250 paths, representing a total of 280 to 40000 numerical simulations, with an average of 500 paths. Both Brunetti et al. (2018) and Turco et al. (2017) combined the Hydrus model (1D and 2D, respectively) with the Morris method. Brunetti et al. (2018) set $r = 100$ for a total of 1700 simulations, and Turco et al. (2017) set $r = 8$ for a total of 40 simulations. In this study we set

$r = 250$, considering that the data will be further analyzed by the Sobol GSA. This gave us a total number of 3000 and 4000 simulations for the analysis with and without clay layers, respectively.

### 2.3. 2 The Sobol method

While the OAT and the Morris sensitivity methods are difference-based, the Sobol-Saltelli method is variance-based (Saltelli and Annoni, 2010). Variance-based methods are used to quantitatively identify both the importance of individual model

parameters and parameter interactions. The Sobol method is based on a decomposition of the total model variance into two main elements: variance of the individual parameter and variance due to interaction with other parameters (Sobol, 2001). Decomposition of the model variance can be written as follow (Saltelli et al., 2004),

$$V = \sum_{i=1}^{k} V_i + \sum_{i=1}^{k}\sum_{j>i}^{k} V_{ij} + \cdots V_{1,2,\ldots,k} \tag{13}$$

$$V_i = V[E(Y|x_i)] \tag{14}$$

$$V_{ij} = V[E(Y|x_i x_j)] - V_i - V_j \tag{15}$$





where $V$ stands for the total variance of the model output. $V_i$ is the variance of each input parameter $x_i$, and $E(Y \mid x_i)$ represents the mean of the system response $Y$ when the parameter $x_i$ is fixed at different values. $V_{ij}$ represents the variance due to interactions between two parameters $x_i$ and $x_j$, and $V_{1\ldots k}$ describes the variance among $k$ parameters. These elements, represented by Sobol's sensitivity indices (SI's), provide quantitative information about the variance associated with a single

parameter or related to interactions of multiple parameters. The main sensitivity index or the first-order sensitivity index $S_i$ quantifies the main effect of parameter $x_i$ on the total variance of $Y$, excluding the interactions with other parameters,

$$S_i = \frac{V_i}{V(Y)} \tag{16}$$

The total-order sensitivity index $S_{Ti}$ of a single parameter $x_i$ includes both the parameter's main variance effect and the proportion of the variance due to interactions of $x_i$ with the other parameters,

$$S_{T_i} = S_i + \sum_{i \neq j} S_{ij} + \cdots + S_{1\ldots k} \tag{17}$$


The values of the indices vary from 0 to 1, where 0 stands for no influence and 1 for a strong influence on the variance. Basically, the calculation of the Sobol indices requires $r(2k + 2)$ model simulations. An increase in the number of $r$ will increase the accuracy of the Sobol indices. Since the number of $r$ is somewhat arbitrary, convergence analysis of the Sobol indices would be the recommended procedure for estimating $r$. However, this approach is time-consuming because it needs

to repeat the GSA multiple times by increasing the number of samples until the variability of the indices between two consecutive analyses is below a certain threshold value for all parameters. The literature reports a very wide range of sample sizes $r$ used in hydrological models, from 7 to 75000 for a total of 300 to $> 6X10^6$ numerical simulations (Song et al., 2015). Wainwright et al. (2013) observed that the sensitivity indices stabilize at around $r = 200$ to 250 (for five parameters). Nossent et al. (2011) used $r = 12000$ in a study testing the suitability of the Sobol GSA for a complex

environmental model (of 26 parameters). They reported that for most parameters, $r < 5000$ was sufficient to reach a stable $S_i$. Brunetti et al. (2016) used $r = 5000$ for a Sobol analysis that used Hydrus 1D model. Based on these observations, a value of $r = 5000$ was chosen for this study.

More details on the Sobol sampling technique and confidence interval establishment can be found in appendix SI.

The Morris and the Sobol sensitivity analyses were executed using the Python programming language and specifically, the

Sensitivity Analysis Library (SALib) (Herman & Usher, 2018). SALib uses a Python script that overwrites the input

parameters given by the GSA in the relevant Hydrus 1D model input files. The script then executes the model and returns the

final aquifer solute concentration and the total solute flux to the aquifer at the end of the simulation. This procedure is

repeated for the number of runs set for the GSA, with changes in the input parameters according to the sampling technique.

SALib then computes the indices of the two methods: $\mu_i^*$ and $\sigma_i$ and their confidence interval for the Morris method, and the

Sobol indices and their confidence interval for the Sobol method. Both Morris and Sobol methods have already been applied

with the Hydrus software package by Périard et al. (2013) (Hydrus - 2D / 3D), Brunetti et al. (2016, 2018) (Hydrus - 1D),

Turco et al. (2017) (Hydrus - 2D), Hartmann et al. (2018) (Hydrus - 2D) and others.

### 2. 4 Treatment of missing output data

Owing to the arbitrary choice of input parameters and multiple model runs, the model sometimes does not converge and

crashes. This creates a problem in analyzing the data because sampling order is important for the analysis results of most

GSA methods. Since there is not yet an agreed and established way to handle these missing data (Sheikholeslami et al.,

2019), we tested the following methods:

**Missing data removal** - by removing the missing data points, or by removing full trajectories, the order of sampling within

trajectories remains undisturbed. Yet, by removing data, valuable information can be lost. In addition, in most methods,

removing data may render the entire sample since it no longer follows the sampling sequence and data structure. Thus, this

study only tested value removal for the Morris method where full trajectories can be removed.

**Missing data imputation** – a missing value is replaced with some other value. The following missing data imputation

approaches were tested:

**Constant value substitution** is an easy and computationally cheap method for the imputation of missing data. The missing

data can be replaced with zeros in cases where the output is typically near zero, or with the mean or the median, in cases

where the distribution is skewed. Sheikholeslami et al. (2019) for example, used the median substitution technique for a





rainfall-runoff model and a land surface hydrology model. A shortcoming of this replacement methods is the potential for

reducing the variance and distorting other statistical properties of the output (Sheikholeslami et al., 2019). In this study, both

the zero and the median substitutions gave similar final GSA indices, with slightly different confidence intervals. Therefore,

only the zero substitution results are presented.

**K Nearest Neighbour (KNN) substitution** - The KNN technique uses neighbourhood observations to fill in missing data.

The underlying rationale behind the KNN-based techniques is that the sample points closer to $x_i$ should provide better

information for imputing the failed output, where $x_i$ is an input parameter vector for which a simulation model fails to return

an outcome. In the KNN method, the failed output is replaced by a response value of a weighted average of the K (the

number of samples) nearest neighbours (KNN). The KNN algorithm computes the distance of the test observation to every

observation in the K nearest neighbours and then imputes the missing value with the average model response of the K

simulations (Duneja and Puyalnithi, 2017). The computed neighbouring distance between the samples is typically the

Euclidean distance (Duneja & Puyalnithi, 2017; Troyanskaya et al., 2001). Lower K values generally result in predictions

with high variance and low bias and vice versa for high K values (Hastie et al., 2009). Thus, in this study, we tested both a

lower K value of five, as was used by Shapiro & Day-Lewis (2022) for groundwater hydrology model, and a higher K equal

to the square root of the size of the data set - a rule of thumb in the KNN method reported to correctly distinguish signal

from noise (Duneja and Puyalnithi, 2017). The KNN analysis was conducted using the programming language Python with

the *scikit-learn* KNN regressor.

**RBF emulation-based substitution** - Model emulation or surrogate modelling, is a strategy that develops statistical, cheap-

to-run surrogates for the output of complex, computationally intensive models (Razavi et al., 2012a). The emulator usually

uses a low computational cost function that fits the non-missing response values $Y_a$ to predict the values for the missing

response $Y_m$. There are various types of model emulations that can be used for hydrological models such as polynomial

regressions, kriging, artificial neural networks, radial basis functions (RBFs), and support vector machines (Razavi et al.,

2012; Zhou et al., 2022). RBF is one of the most commonly used function approximation techniques, because it can provide

an accurate emulation of high-dimensional problems for low computational cost. Sheikholeslami et al. (2019), for example,



employed the RBF approximation for crashed model simulation emulation which performed better than all other methods tested in that study. The RBF approximation is a weighted summation of $n_a$ number of functions that can approximate the predictive response $Y$ at a point $x_i$. Here $n_a$ was set to the number of non-missing sample points. Detailed equations of the RBF approximation can be found in the appendix SI. The RBF imputation analysis was also conducted with the Python

program using the SciPy RBF interpolation package.

## 3.    Results & discussion

In the following sections, we present the results of the Morris and Sobol analyses for benzene transport in the unsaturated zone mimicking Israel's coastal plain vadoze zone. The analyses were initially conducted for a homogenous media of sandy soil and later for a heterogeneous media containing clay layers. Thus, the results first present the simpler Morris analysis for

the homogenous media, then the Sobol analysis for the same media, and finally the Morris analysis results for the heterogeneous media.

### 3. 1 Homogenous media analyses

#### 3.2.1 Morris analysis for homogenous sandy soil

In this analysis, the model's sensitivity to seven input parameters ($k$) was tested (Table 1), considering two model outputs:

benzene aquifer concentration at the end of the simulation and benzene cumulative flux to the aquifer. The analysis was conducted for 250 paths ($r$) and six levels ($p$; i.e., dividing the parameters' space to five equal segments, where the parameter can be assigned six different values) for 2000 simulations overall ($r\,(k\,+\,1)$). Out of these simulations, only 42 simulations (2.1 %) did not converge or crashed. To avoid bias in the results, we used the methodology described in Section 2.4 to either replace the missing values or remove the trajectories that contain missing values. Three imputation methods are presented:

zero substitution, the RBF emulator, and the KNN method with K = 5 and K =  45 (representing the square root of the sample size). Figure 2 presents $\mu^*$ for these different methods. Detailed values of all indices for the different methods can be found in the appendix (Tables S3 - S4). Small differences were observed between the different methods for $\mu^*$, $\mu^*$





confidence interval, and $\sigma$ values for each of the input parameters, with the same order of parameter importance. The similarity between the different methods stems from the scarce missing values, hardly affecting the overall results. In all

strategies for handling missing data, it is evident that the GSA performed the worst for the weakly influential parameters - $\alpha$, $K_s$ and $\alpha_l$, exhibiting a high ratio of $\mu^*$ to $\mu^*$ confidence interval (Fig. 2, Tables S3 - S4) this was also evident in GSAs obtained with the Hydrus model for other hydrological problems (Brunetti et al., 2016, 2022; Hartmann et al., 2018; Zhou et al., 2022; Brunetti et al., 2017).

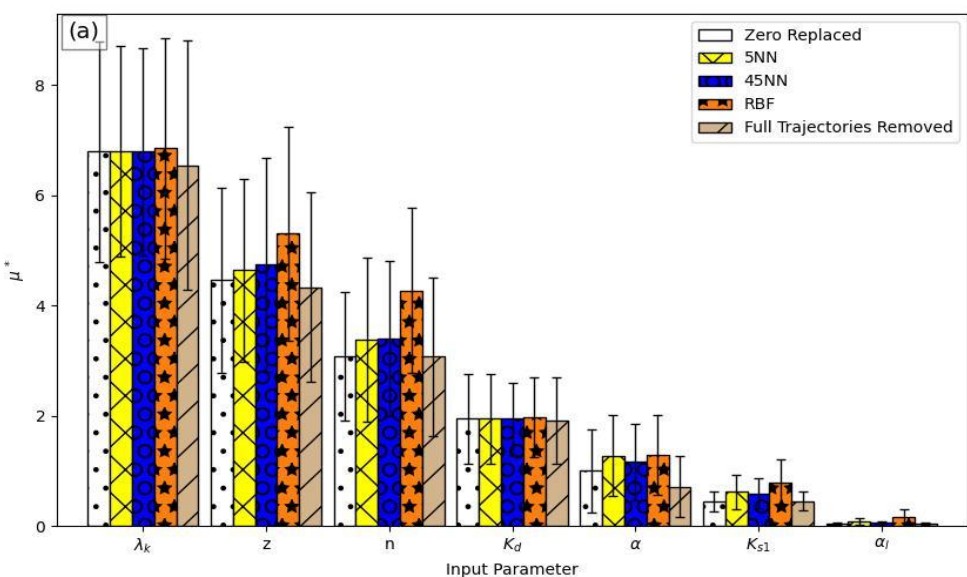



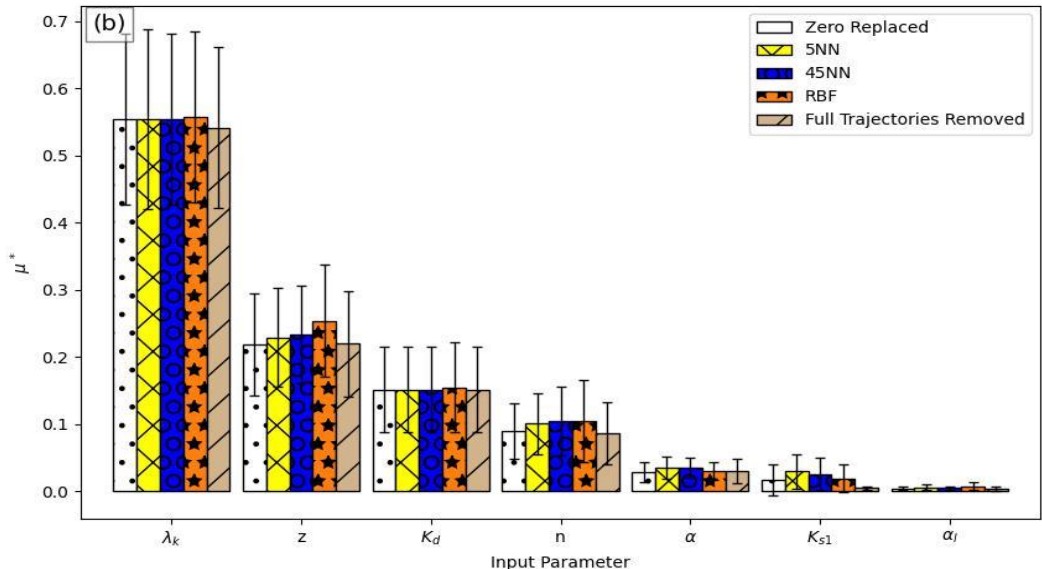


**Figure 2 – Morris analysis results for homogenous sandy soil obtained with the different methods for missing data imputation/removal for: a. Cumulative benzene flux to the aquifer and b. Final benzene concentration in the aquifer. Black bars represent $\mu^*$ confidence intervals.**

The effect of the different parameters on the output can also be seen in Figures 3a and 3b, where $\mu^*$ versus $\sigma$ is presented for

the Morris analysis conducted with the RBF emulation substitution method used to replace the crashed data. Though

minimal differences were observed between all methods (Fig. 2), the RBF results are shown here for consistency purposes,

since the RBF method gave the best results in the heterogeneous media case (see Section 3.2.2).

In all cases, the input parameter with the strongest effect on the system was the degradation rate $\lambda_k$, followed by the profile

depth $z$ (Figs. 2 and 3). The next two influential parameters are the adsorption coefficient $K_d$ and the van Genuchten $n$

parameter. Both showed a similar effect on the concentration, though the effect of $n$ on the flux to the aquifer was much

more pronounced. Finally, the van Genuchten $\alpha$ parameter, the hydraulic conductivity $K_s$ and the dispersivity $\alpha_l$, showed

little effect on the model results.





Zanello et al. (2021) reported similar results in a LSA for a model of BTEX transport in an unsaturated homogenoues sandy

soil using Hydrus 2D/3D software. They found that the order of the input parameters' influence on BTEX arrival to the

aquifer (tested as concentration) was $\lambda_k > K_d > z > K_s$. The stronger influence of $K_d$ compared to $z$ in that study is probably

the result of the low $z$ values tested there (2.5 - 4 m), representing a shallow aquifer. In another study, Davis et al. (1994)

modeled the constant leakage of benzene in a loamy sand soil to an aquifer beneath a manufacturing facility. Benzene

concentrations of ~1 mg/l were found in the groundwater beneath the source (~25 mg/l), though in monitoring wells ~100 m

from the source no bezene was detected. In their LSA they too found that $\lambda_k$ was the "dominant mechanism" for benzene

attenuation and found $K_d$ to be very influential. Moreover, similarly to our study, their model was insensitive to $\alpha_{l,}$ (Davis et

al., 1994). Indeed, the great importance of biodegradation for the removal of gasoline hydrocarbons in aerobic environments

has been recognized and reported in the literature (Lahvis et al., 1999; Berlin et al., 2016; Yadav and Hassanizadeh, 2011;

Mohanadhas and Kumar, 2019; Alvarez et al., 1991; Abu Hamed et al., 2004), here it is demonstrated once again.

Generally, the order of influence of the parameters was similar for the cumulative flux to the aquifer and for the final

concentration in the aquifer (Fig. 3). For both the flux and the concentration, a correlation between $\mu^*$ and $\sigma$ is observed, as

reflected in their arrangement around the same diagonal line (Fig. 3, red line), indicating that none of the parameters have

solely a linear effect ($\mu^*$ being the mean effect). Instead, all parameters exhibit an interaction effect ($\sigma$ - the standard

deviation of the effect), where the interactions increases with the increase in the mean.

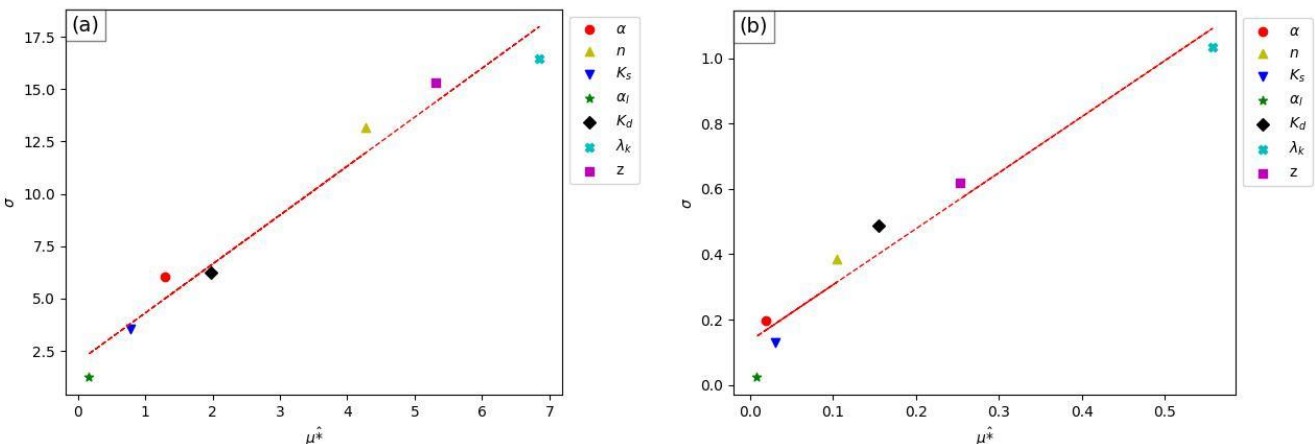

**Figure 3 - Morris analysis results for homogenous sandy soil with RBF imputation for: A. Cumulative benzene flux to the aquifer and B. Final benzene concentration in the aquifer.**

### 3.2.2. Sobol analysis for homogenous sandy soil:

The Sobol analysis for homogenous media was conducted for the four most influential parameters of the Morris analysis: $\lambda_k$, $z$, $n$, and $K_d$, to obtain more quantitative information on the parameters' influence and interactions. Five thousand sets of parameters were generated, constituting an overall total of 50000 model runs for $r\ (2k + 2)$. Of the 50000 model runs, 881 samples did not converge. The same methodology used for the Morris analysis was used for missing values imputation (Section 2.4). Yet, with the Sobol analysis, it was impossible to remove the missing points, since the order of sampling is significant to the overall analysis, and sampling is not divided into sets of trajectories. Hence, the data removal method was not used.

Figures 4 and 5 show the $S_1$ and $S_T$ Sobol indices for the different methods of missing data replacement. Detailed and averaged values of all methods can be found in the appendix (Tables S5 - S10). All missing data imputation methods gave similar results (Figs. 4 and 5, and Tables 5 and 6). This is expected given the low dimensionality of the model (four input parameters). Here, the GSA also performed the worst for the weakly influential parameters ($n$ and $K_d$) exhibiting a high confidence interval to indices ratio.





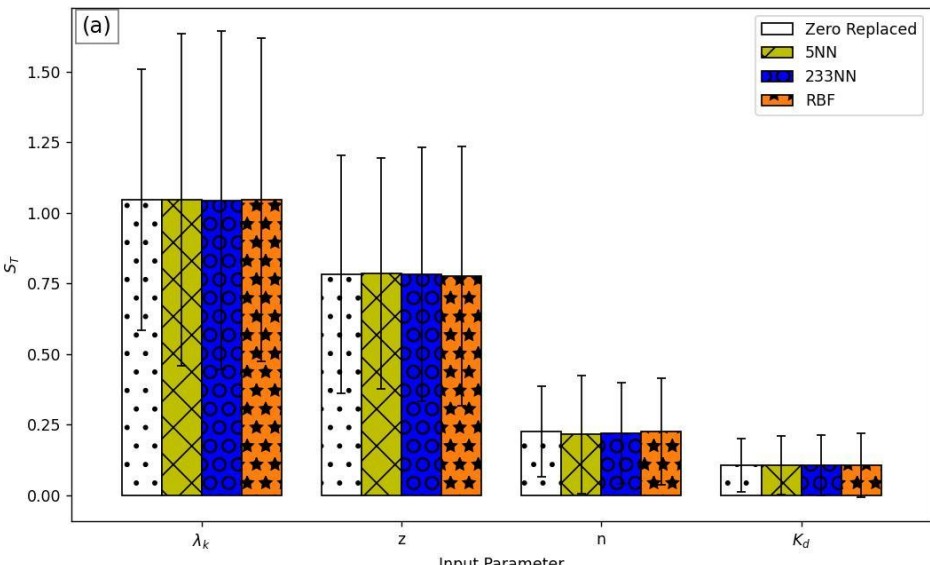


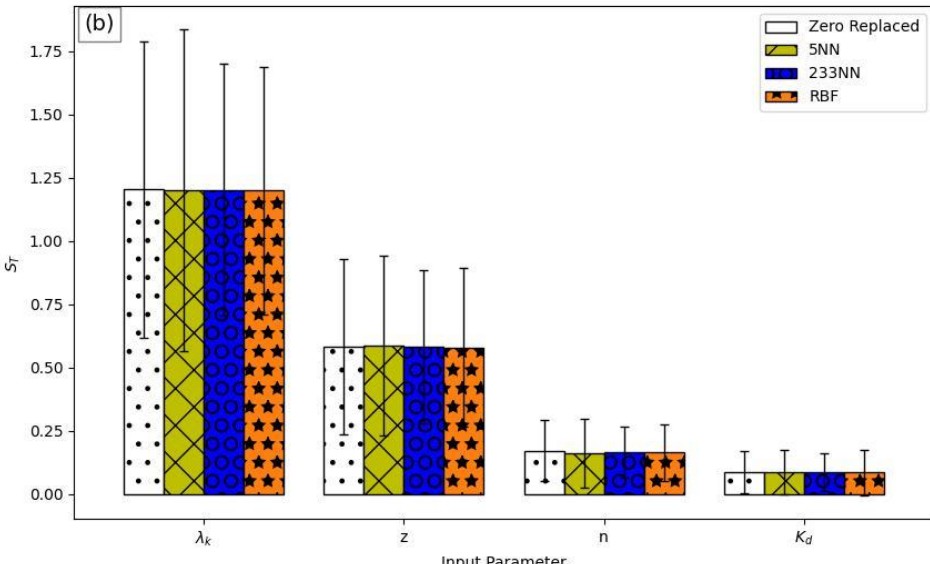

**Figure 4 – The Sobol total indices ($S_T$) for homogenous sandy soil obtained using different methods for missing data imputation for: a. Cumulative benzene flux to the aquifer and b. Final benzene concentration in the aquifer. Black bars represent $S_T$ confidence intervals.**



A similar effect of the different parameters' order and magnitude of importance on the two outputs was observed. Just like

the Morris analysis, here as well, $\lambda_k$ and $z$ were found to be the most influential parameters with the highest total order ($S_T$)

and first order ($S_1$) indices values (Figs. 4 and 5). $S_T$, unlike $S_1$, often sums to more than 100% because it is the sum of $S_1$ and

all the higher-order Sobol indices involving the parameter (Saltelli et al., 2004). The difference $S_T - S_1$ is a measure of how

much parameter $x_i$ is involved in interactions with any other input variable (Saltelli et al., 2004). The total index $S_T$ (Figs. 4

and 5) demonstrates that most of the variance in both flux and concentration is caused by $\lambda_k$, consisting of the variation of $\lambda_k$

itself ($S_1$ of ~11.38 % and 13.21 % for the flux and concentration, respectively, Tables S5 and S6) and the interactions with

other parameters. It should be noted that $z$ has a relatively low main effect ($S_1$ of 1.85 % and 1.17 % for the flux and

concentration, respectively, Tables S5 and S6) but a high total effect of ~58 % and 78 % for the flux and concentration,

respectively (Tables S7 and S8), indicating that this parameter has a limited direct impact on the variance of the output, but a

strong interaction effect, most likely with the degradation coefficient. The total Sobol' index of an input parameter is the sum

of the first-order Sobol' index *and* all the higher-order Sobol' indices involving that parameter. Hence, The sum of the total

Sobol sensitivity indices is equal to or greater than one (Gatel et al., 2020). If no higher order interactions are present, the

sum of both the first and total order Sobol indices are equal to one. Sum of $S_T$ values >100% was also reported by Brunetti et

al. (2017), Schübl et al. (2022), Zhou et al. (2022) and (Nossent et al., 2011).

Ciriello et al. (2017) performed a Sobol analysis for benzene contamination in an unsaturated soil assuming very deep

aquifer where contamination will not arrive, and in a shallow aquifer. They reported $K_s$ as one of the most important

parameters, while $\alpha$ and $n$ were both found mostly insignificant. Yet, it is hard to compare between that study and this one

because $\lambda_k$, $K_d$ and $z$ that were found here highly significant, were not tested in that study as well as moderate aquifer depths

(more than 5 m) that were tested here.


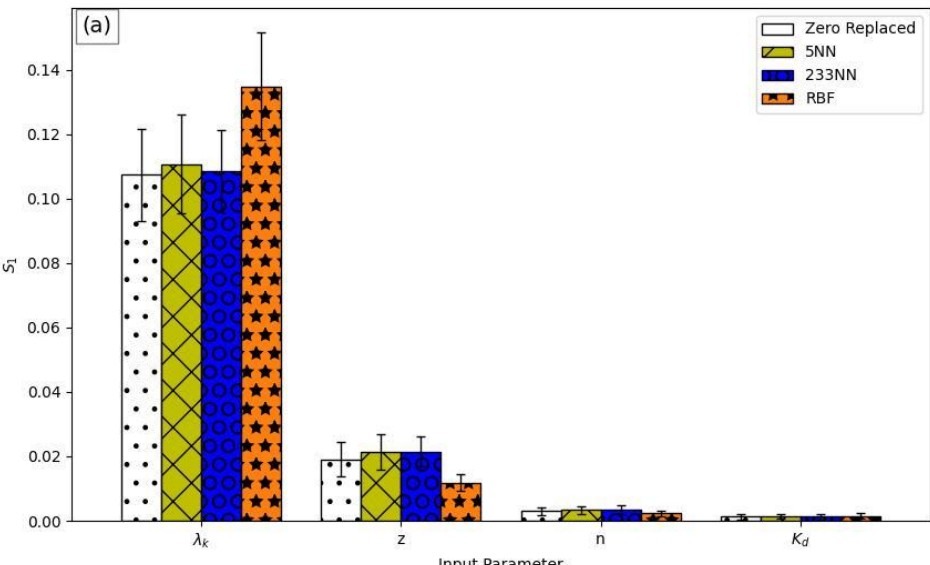

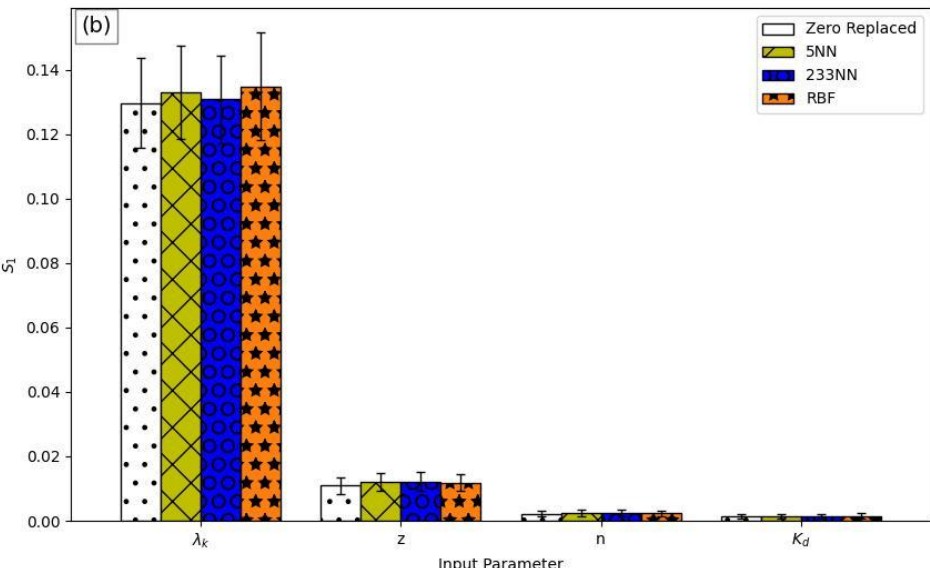

**Figure 5 – Sobol first order indices (S₁) for homogenous sandy soil obtained using different methods for missing data imputation for: a. Cumulative benzene flux to the aquifer and b. Final benzene concentration in the aquifer. Black bars represent μ\* confidence intervals.**



$\lambda_k$ is the only parameter with $S_1$ higher than 10 %, hence the only parameter with a strong main effect on the output variance. When the sum of all first order indices is less than 100 %, the model is non-additive, meaning that it is affected by interactions (Neumann, 2012; Nossent et al., 2011). Here, the sum of all first-order indices is < 15 %, indicating that the model is non-additive and very much affected by interactions. Only 13.7 % and 14.8 % of the variance for the flux and concentration, respectively, are attributable to the first-order effect (Tables S5 and S6; showing the sum of $S_1$ for the flux and

concentration, respectively), highlighting the fundamental role of interactions between parameters.

Overall, the Sobol method results agreed with those of the Morris method. Indeed, the Morris method was proposed as an efficient tool to be used prior to variance-based GSA, in order to screen important and unimportant factors and to provide the first inspection of the model's behaviour at a reasonable computational cost (Brunetti et al., 2018; Song et al., 2015; Wainwright et al., 2013). Similarity between Morris and variance-based methods was also observed by Herman et al. (2013)

and Sarrazin et al. (2016).

### 3. 2 Heterogeneous media

The following analysis included clay layers within the sandy soil to obtain a more realistic representation of Israel's coastal plain vadose zone, mostly comprising sandy soil but also including clay layers and interbeds (Ecker, 1999).

To create a representative configuration of the clay layers in the vadoze zone above the Israel's coastal plain aquifer, we

examined the distribution of clay layers in selected fuel-contaminated sites. For that purpose, we constructed a database consisting of records from 32 fuel-contaminated sites containing dozens of monitoring boreholes that were obtained from the Israel Ministry of Environmental Protection. Each borehole in the database was sampled at multiple depths and characterized for the soil type. We classified these soil types into four main categories: gravel, sand, clayey sand (consisting of 55 % sand and 45 % clay), and clay, according to the soil type name on the database (see Table S9 in the appendix for the categories).

For each site, the percentages of each of the four soil types at each specific depth was extracted (i.e., the number of boreholes having a given soil type at a specific depth divided by the total number of boreholes penetrating that depth). We then looked for the percentage of clay at each depth, where a layer having clay percentage higher than 25 % was considered a clayey





layer (Whiting et al., 2011). This yielded the number and thicknesses of clay layers for each site (an example of one site can

be found Table S10 in the appendix). Based on this methodology, a distribution of clay layers in contaminated sites at

Israel's coastal plain vadose zone was calculated and is presented in Table 3.

**Table 3 – Summary of clay layers distribution**

| Number of clay layers | Number of sites | Percentage of sites | Thickness of clay layers (m) | | | | |
|---|---|---|---|---|---|---|---|
| | | | Mean | STDEV | Max | Min | Median |
| 0 | 1 | 3.13 % | -- | -- | -- | -- | -- |
| 1 | 16 | 50 % | 3.8 | 3.21 | 11.7 | 0.1 | 3 |
| 2 | 4 | 12.50 % | 2.38 | 2.26 | 8.1 | 0.2 | 2 |
| 3 | 5 | 15.63 % | 1.135 | 0.94 | 3 | 0.1 | 1.95 |
| 4 | 2 | 6.25 % | 2.3 | 2.78 | 6.4 | 0.3 | 1 |
| 5 | 2 | 6.25 % | 4.2 | 6.76 | 12 | 0.1 | 0.5 |
| 6 | 2 | 6.25 % | 2.55 | 3.46 | 5 | 0.1 | 2.55 |

Table 3 shows that only in 1 of the 32 examined sites, there were no clay layers at all. Fifty percent of the sites had only one

clay layer and most of the sites had 1-3 layers (~78 %), whereas almost 20 % had 4 - 6 layers. The mean of the layers'

thickness ranged from 1-4 m. Yet, the standard deviation was high and the actual thickness ranged from 11.7 to 0.1 m. Due

to this variance in the distribution of clay layers (Table 7), it was decided to examine the number of clay layers ($N$) and their

thickness ($b$) as additional input parameters in the sensitivity analysis of the heterogeneous media within the range of values

reported in Table 3. The range of tested $N$ and $b$ can be found in Table 2. We are not aware of other studies that tested the

distribution of clay layers in a SA for contaminants transport.





### 3.2.1 Morris analysis for heterogeneous media:

In this analysis, 12 parameters concerning the soil type were examined: $\alpha$, $n$, $K_s$, $\alpha_l$, $\lambda_k$ and $K_d$, both for sand and clayey soil (represented below with a subscript of 1 and 2, respectively). Three additional general profile parameters were tested: $z$, $N$ and $b$, comprising 15 parameters overall (Table 2).

The analysis was conducted for 250 paths ($r$) and four levels for an overall total of 4000 simulations ($r\,(k+1)$), from which 338 did not converge or crashed. The increase in the ratio of failures compared with the previous analysis (2.1 % for the homogeneous Morris analysis versus 8.45 % here) can be attributed to the complex transport in the heterogeneous medium, and the difficulty in modeling flow between sand and clay layers, as well as to the increase in the number of model parameters (dimensionality of the parameter space) increasing the arbitrary combinations of parameters during GSA (Sheikholeslami et al., 2019). The same methodology was used for missing data imputation or removal, as discussed above.

Unlike the previous analyses, the different methods for missing values imputation yielded dissimilar confidence interval levels, as well as dissimilar $\mu^*$ values for some parameters (Fig. 6). Also, an overall increase in the ratio between $\mu^*$ and its confidence interval was observed (Fig. 6). Again, the GSA performed the poorest for the less influential parameters, exhibiting the highest ratio of confidence interval to $\mu^*$, as evident in Figure 6 for the parameters on the right side of the charts. In the appendix, the ratio between $\mu^*$ confidence interval to $\mu^*$ is presented in Fig. S1 for a clearer view on the diffrence between the different methods and parameters. GSA results with high confidence interval values were also reported by other studies that used Hydrus models (Brunetti et al., 2016, 2022; Hartmann et al., 2018; Zhou et al., 2022; Brunetti et al., 2017). Though the authors do not adress this issue, it indicates the need for more model runs to obtain convergence of the indices (Sarrazin et al., 2016). Yet, a clear $\mu^*$ ranking is observed with an overall consistency with the previous results of the homogeneuous case and between the different methods.

Full trajectory removal performed the poorest for most parameters, while the RBF emulation method performed the best (Fig. 6 and Fig. S1). The KNN method gave better results than the zero substitution, especially when the effect on the concentration was examined (Fig. 6b). For the concentration, the 45NN performed better than the 5NN. Differences between





the two KNN methods were less pronounced for the cumulative flux. These results are similar to those reported by Sheikholeslami et al. (2019), where the RBF emulation-based substitution performed better than the single NN and than a

constant value substitution (the median, in their case). Detailed values of all methods indices can be found in the appendix Tables S11 - S12. A correlation between $\mu^*$ and $\sigma$ for all input parameters is again demonstrated by their arrangement around one diagonal line (the red line in Figures 7a and 7b), indicating that the interactions' effect increases with the increase in the total effect of each parameter.

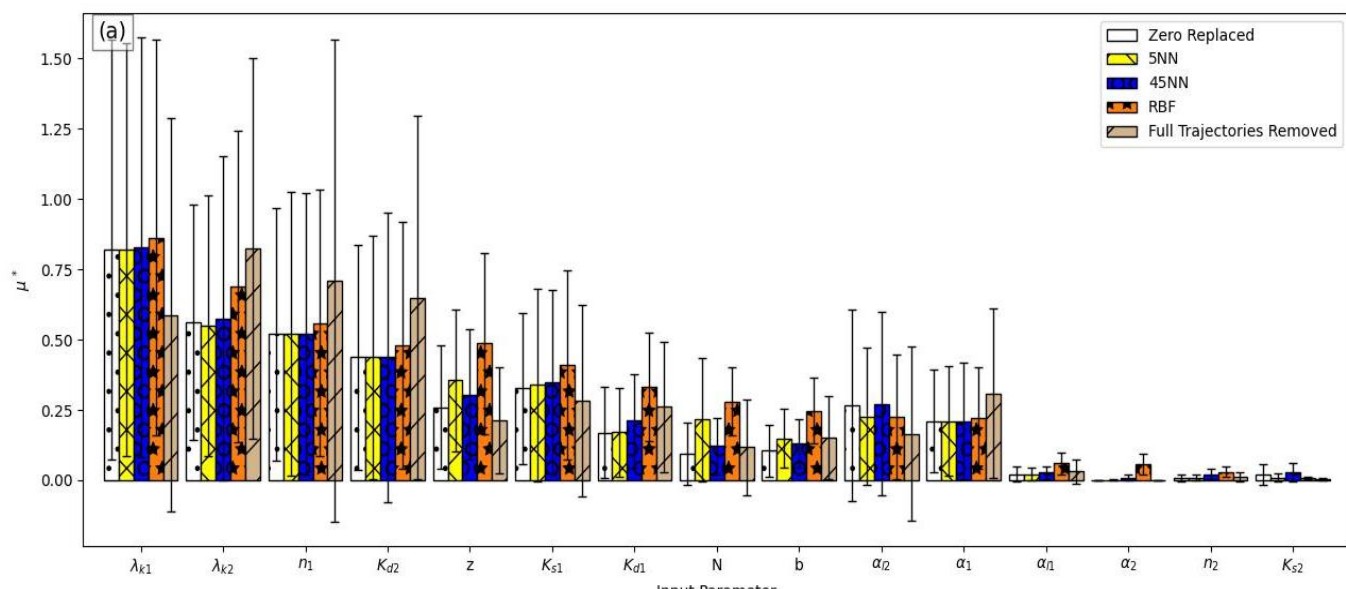

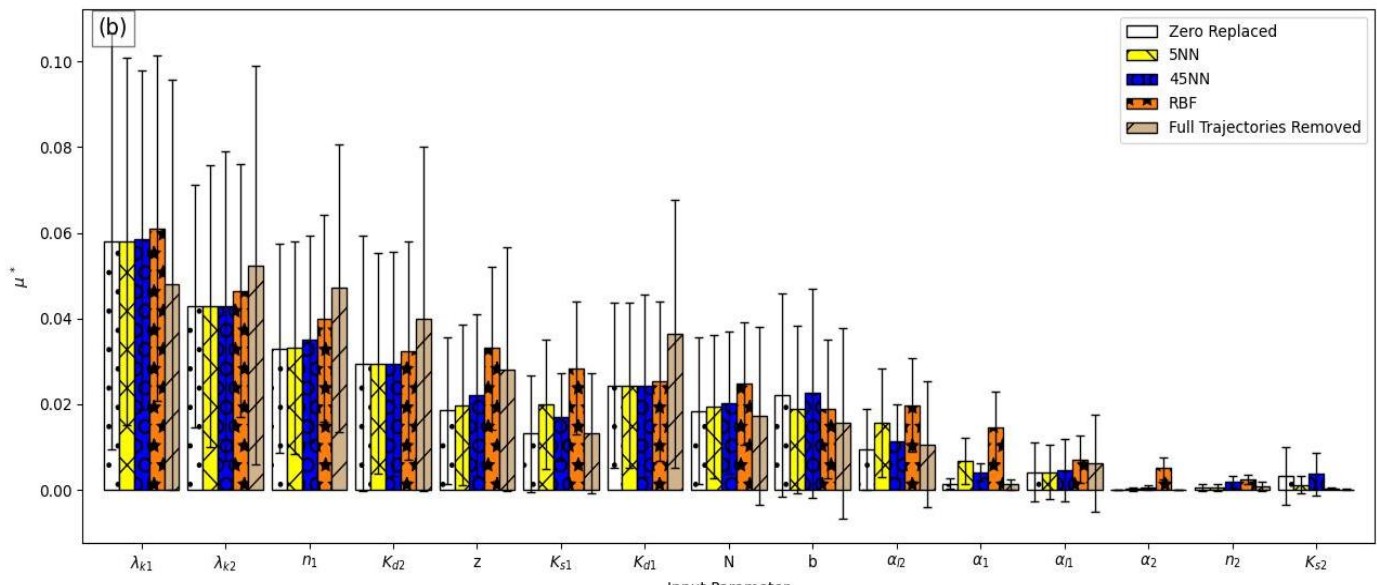


**Figure 6 – Morris analysis results for heterogeneous media, μ$^{*}$ values for: a. cumulative benzene flux to the aquifer; b. final benzene concentration in the aquifer.**

In Figures 7a and 7b, the effect of the different parameters on the outputs, in terms of $\mu^*$ versus $\sigma$, when using the RBF method, is shown. Like the GSA results of the homogenous media, it is evident that the degradation coefficients of both soils and the van Genuchten $n$ parameter of the sand layers $n_1$ are the most dominant factors controlling benzene transport to the aquifer. The degradation coefficient of the sand layers $\lambda_{k1}$ was found to be the most dominant parameter, considering both the cumulative flux and the concentration at the end of the simulation. Next, the degradation coefficient of the clay layers $\lambda_{k2}$ and $n_1$ share the second and third places of influence. While $n_1$ was found to be more influential for the flux, $\lambda_{k2}$ had a stronger effect on the concentration. This makes sense since $n$ primarily affects water flow, whereas $\lambda_{k2}$ affects benzene directly. In the Morris homogenous analysis $n$ also affected the flux more than the concentration. Therefore, although the two parameters are substantial, we see that the relative effect also depends on the output tested. We note that $n_1$ was also third/fourth in significance for homogenous soil analysis; thus, it is a very significant parameter for modeling the transport of





benzene. In the fourth place here, the clay adsorption coefficient $K_{d2}$ was found to be equally influential for both outputs, exhibiting an increased interaction effect (high $\sigma$). Benzene adsorption for clay materials is higher than that of sand
(Mohanadhas and Kumar, 2019; Zytner, 1994), and was set accordingly in the GSA (Table 1); therefore, its increased influence is not surprising.

Following these four very influential parameters, we see a large group of moderately influential parameters. For both outputs, the sand's saturated hydraulic conductivity $K_{s1}$ and the number of clay layers $N$ are close together in the fifth and sixth place for the flux, and seventh and eighth place for the concentration. In both cases $K_{s1}$ shows a higher total effect ($\mu^*$)
and a smaller interaction effect ($\sigma$) than $N$. For the flux, $K_{s1}$ and $N$ are more influential than the profile depth $z$ positioned in seventh place, whereas for the concentration, it's the other way around and $z$ occupies the fifth place in importance. Compared to the homogenous sandy media, $z$ moved downward in order of importance, mostly due to the increased influence of the clay layers in retardation of benzene, as manifested by the high importance of clay parameters like $\lambda_{k2}$, $K_{d2}$, and $N$. Contradictory to $z$, $K_{s1}$ moved upward in the rank of importance. Since sand comprises most of the profile and the
movement in the sand is faster than in the clay layers, this parameter now plays a significant role, especially for the flux. The number of clay layers $N$ moderately affects both outputs, while the layers' thickness $b$ also moderately affects the concentration, but is somewhat less important for the flux. The adsorption coefficient of sand $K_{d1}$ is moderately-high in importance (seventh place) for concentration, whereas for the flux it is only at the 11[th] place. A similar trend was observed in the homogenous media analysis where $K_{d1}$ had a stronger effect on the concentration, probably since sorption lowers
benzene concentration but has a lower effect on total flux.





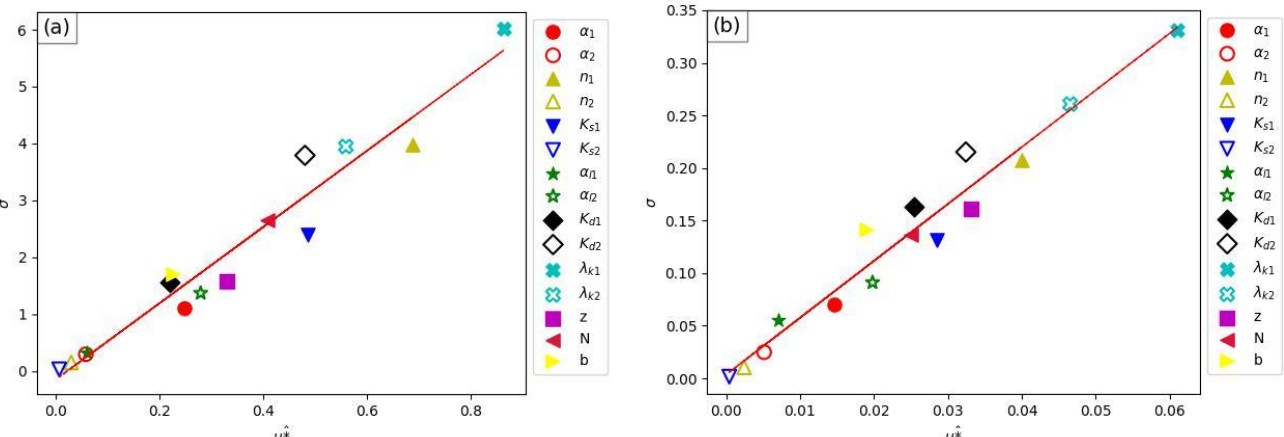

**Figure 7 – Morris analysis results for heterogeneous media for: a. Cumulative benzene flux to the aquifer and b. Final benzene concentration in the aquifer.**

Following these influential parameters, the clay layers' dispersivity $\alpha_{l2}$, and $\alpha_1$ of the sand layers, are less influential but still

somewhat close to the middle of the graph. The least influential parameters are mostly soil properties of the clay layers, such

as the clay hydraulic conductivity $K_{s2}$, $n_2$ and $\alpha_2$ parameters, and the dispersivity of the sand layers $\alpha_{l1}$. Interestingly, Gribb

et al. (2002), who conducted LSA for a risk assessment model of benzene and naphtelene transport to groundwater through

sand, loam, and clay soils, also reported high model sensitivity to $\lambda_k$ and $K_d$ for all soils. In their case, the model was also

less sensitive to $K_s$, except for pure loam and clay soils. For other parameters that were not tested here (porosity, bulk

density, residual water content and initial concentration), the model was only sensitive in the case of pure clayey soil. Yet,

their study assumed homogenous media of each soil type. Here, most clay parameters are less influential, probably due to the

smaller fraction of clay, as compared with the sand layers, which comprise most of the profile.

The results for both the homogenous and heterogeneous media showed that the most dominant factor controlling benzene

arrival to the aquifer is $\lambda_k$, especially in the sand layers, which occupy most of the profile. Since the values of $\lambda_k$ vary

greatly in the literature, a careful examination and selection of this parameter is recommended in hydrological modeling for

benzene transport, and further research to elucidate its value onsite is encourged.

Following $\lambda_k$, $n_1$ was found to be highly dominant in this analysis, whereas in the homogenous media $z$ was more significant. The aquifer depth is an easy-to-measure parameter, and it should be included in any model for benzene transport. Also, $n$ can be established quite easily using tools like Rosetta to establish soil texture (Schaap et al., 2001). Therefore,

examination of soil types onsite is also extremely important. The adsorption coefficient of the clay layers was also found to be highly significant as well as the number of clay layers. Most studies that tested SA for benzene transport in different soils types, used homogenous media representation of each soil type and tested one soil type at the time (Davis et al., 1994; Gatel et al., 2019; Gribb et al., 2002). Here, we test the effects of both the parameters of the individual soil types and the distribution of the clay layers through $b$ and $N$. This provides a better assessment of the importance of each parameter of

each soil type, and shows that the representation of clay layers distribution should be considered carefully. Therefore, the occurrence and number of clay layers should be characterized and considered when examining contaminated sites, even for the occurrence of thin layers. On the other hand, non-influential parameters may be fixed to literature values. Those parameters include soil properties of the clay layers, such as the clay hydraulic conductivity $K_{s2}$ and $n_2$, as well as parameters like the dispersivity $\alpha_l$ and van Genuchten's $\alpha$ parameter of both layers.

**4.   Summary & conclusions**

This paper explores the effect of different model parameters on benzene transport in the vadose zone of Israel's coastal plain aquifer and its potential arrival to the aquifer below. A physical model was implemented to simulate benzene transport in the unsaturated zone. The model was initially employed for homogenous sandy soil, as sand comprises the vast majority of the vadose zone. Next, the model was set to describe heterogeneous soil containing clay layers representing lithology obtained

from data of contaminated sites. Two GSA methods were applied to examine the effect of the model input parameters on benzene concentration in the aquifer at the end of the simulation, and on benzene cumulative flux to the aquifer. Additionally, treatment of missing data due to model crashes was demonstrated.

The results for both the homogenous and heterogeneous media showed that the most dominant factor controlling benzene arrival to the aquifer is benzene degradation coefficient ($\lambda_k$), especially in the sand layers which occupy most of the profile.

Following $\lambda_k$, van Genuchten $n$ parameter was found to be highly dominant, mainly in the heterogeneous media, whereas in the homogenous media the depth ($z$) was more significant. The adsorption coefficient of the clay layers and the number of clay layers were also found highly significant.

A substantial interaction effect between the parameters was observed, where the parameters with the highest individual effect showed a high interaction effect and vice versa. The degree of individual parameter influence on the model was shown to be

small ($< 15\ \%$) by the Sobol analysis, indicating the great importance of interactions between parameters.

The different methods for missing data handling yielded a similar overall ranking of the influential parameters identified by the GSA. However, the RBF emulation-based substitution showed better results compared to the KNN and zero substitution techniques, particularly when the transport between layers was considered, and the model dimensionality and subsequent number of failures was high. In that case, the data removal technique performed markedly worst. Last, it was observed that

the GSA and different methods for data imputation performed the best for the more influential parameters.

## 5. Author contribution

RR and NS planned the research; MC ran the model and GSA and analyzed the data; MC wrote the manuscript draft; RR and NS reviewed and edited the manuscript.

## 6. Competing interests

The contact author has declared that none of the authors has any competing interests

## 7. Acknowledgments

This work was funded by Israel Water Authority.

The data set in available online at https://doi.org/10.6084/m9.figshare.22012718.v1





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
