# Peer review of "Identification of Parameter Importance for Benzene Transport in the Unsaturated Zone Using Global Sensitivity Analysis"

_Hydrology and Earth System Sciences, 2023_

## Referee Comment (RC1)

Report on the manuscript "Identification of Parameter Importance for Benzene Transport in the Unsaturated Zone Using Global Sensitivity Analysis" submitted to *Hydrology and Earth System Sciences.*

It is my pleasure to review the manuscript entitled "Identification of Parameter Importance for Benzene Transport in the Unsaturated Zone Using Global Sensitivity Analysis". How to accurately quantify the parameter importance in the complex contaminant transport models has always been an important topic in groundwater research. In this study, GSA methods of Morris and Sobol were implemented to investigate the important parameter for benzene transport model in the unsaturated zone.

I believe this paper is well written with high quality and good logic. I would be in favor of publication after the author addressed the comments given below.

**Major Comments:**

1. In the section 2.3.2, I believe authors should provide more details about the calculation process and algorithm implemented for the Sobol indices. And I believe it is too obvious that the sample sizes were dramatically different for different problems since they were calculating different variances based on different models, it is unnecessary to list these different size numbers.

2. I am not sure why the authors focused on the crashed simulations; it seems to be the problem of numerical model instead of sensitivity analysis problem. And the differences of GSA results using different methods to fulfill these "bad" input samples are more like pure numerical fluctuations to me.

3. In the section 3.2, have the authors tried the geostatistics tools to build different samples of heterogeneous media structure? I don't understand how the different samples representing the vadose zone media were generated. For the sentence in line 464, I believe Dai et al., (2017) has done some similar work and please check if it is helpful to improve this research for the heterogeneous media structure generation through geostatistics implementation.

**Minor Comments:**

1. Line 84-85: I don't think these reference papers are all focus on the sensitivity analysis for the unsaturated zone.

2. Line 90: I don't understand the term "properties of benzene itself", what properties?

3. Line 99-100: I believe authors should provide more details about these two methods, especially about their algorithms, in the introduction.

4. Line 103-104: please provide some references, I don't know this is a common problem.

5. Some formats of titles of subsections are incorrect or inconsistent, please check all of them.

**Reference**

**H. Dai**,X. Chen*, M. Ye, X. Song, and J. M. Zachara, A Geostatistics-Informed Hierarchical Sensitivity Analysis Method for Complex Groundwater Flow and Transport Modeling, *Water Resources Research*, 2017, 53(5), 4327-4343.

---

## Author Response (AR1)

**Response to editor and reviewers comments for the manuscript "Identification of Parameter Importance for Benzene Transport in the Unsaturated Zone Using Global Sensitivity Analysis" submitted to *Hydrology and Earth System Sciences* (all authors' answers are in blue font, all editor's and reviewers' comments are in black font).**

**Editor comments:**

On the basis of the Reviewers' comments and the Authors' replies to these, I do think the manuscript can benefit from a set of revisions that can range between moderate to major. These include technical as well as presentation quality aspects, as elucidated in the Reviewers' reports. It is not my intention to discount any of the constructive comments raised by the Reviewers during the review phase.

We thank you and the two reviewers very much. We have fully addressed the editor's and reviewers' comments and made the relevant changes to the manuscript. Our response is given below in blue.

Additionally, while the GSA strategy addressed by the Authors is broadly used, it would be beneficial to the reader if the Authors could address reasons underpinning their choice to rely on variance-based indices and not on other indices (e.g., moment-based global sensitivity indices, where quantification of uncertainty is not limited to the variance and consider moments of various orders of the probability distribution of the model output of interest).

Variance based methods assume that the variance (i.e. the second moment) is sufficient to describe the output variability. This is not always the case, especially when the model is highly skewed. Hence recently, moment independent methods and moment-based methods were suggested as alternatives relying on the probability density function or a combination of both, where several of the outputs moments are tested. Yet when the different methods were compared, highly influential parameters were usually common, and the ranking of these parameters was similar (Wang and Solomatine, 2019; Upreti et al., 2020; Khorashadi Zadeh et al., 2017), and in some cases variance-based methods were preferred (Upreti et al., 2020). Variance based GSA methods are still the most commonly used since they are conceptually simple and easy to implement. This is why for this study we chose to apply the 'Sobol' method which is still probably the most well-established and widely-applied GSA method.

Therefore the following text was added to the (lines 103-114 of the revised manuscript):

Variance based GSA methods are most commonly used since they are conceptually simple and easy to implement (Upreti et al., 2020; Khorashadi Zadeh et al., 2017; Jaxa-Rozen et al., 2021; Saltelli et al., 2010; Sobol, 2001; Song et al., 2015; Saltelli et al., 2004; Nossent et al., 2011; Brunetti et al., 2016, 2017). Yet, when the model output is highly-skewed or multi-modal the variance may not adequately represent output uncertainty (Liu et al., 2006; Borgonovo, 2007). Therefore alternative methods like moment-independent (Liu et al., 2006; Borgonovo, 2007) and moment-based (Dell'Oca et al., 2017) methods were developed using the output probability density function (PDF) to fully characterize the output uncertainty. In some studies PDF methods were shown to perform better for parameters' importance ranking, though highly influential parameters were usually common, and the ranking of these parameters was similar (Wang and Solomatine, 2019; Upreti et al., 2020; Khorashadi Zadeh et al., 2017), and in some cases variance-based methods were preferred (Upreti et al., 2020).

And also when discussing the specific methods chosen (Morris and Sobol) we added the following (lines 152-154 of the revised manuscript **in bold here**): "the Sobol method (Sobol, 2001), a computationally heavy, variance-based **and probably the most well-established and widely-applied GSA (Khorashadi Zadeh et al., 2017; Jaxa-Rozen et al., 2021; Saltelli et al., 2010; Sobol, 2001; Song et al., 2015; Saltelli et al., 2004; Nossent et al., 2011; Brunetti et al., 2016, 2017, and many more)."**

The manuscript will then be subject to an additional round of reviews, possibly by the same Reviewers who served during this first phase.

**Reviewer #1 comments:**

It is my pleasure to review the manuscript entitled "Identification of Parameter Importance for Benzene Transport in the Unsaturated Zone Using Global Sensitivity Analysis". How to accurately quantify the parameter importance in the complex contaminant transport models has always been an important topic in groundwater research. In this study, GSA methods of Morris and Sobol were implemented to investigate the important parameter for benzene transport model in the unsaturated zone.

I believe this paper is well written with high quality and good logic. I would be in favor of publication after the author addressed the comments given below.

**Thank you for the positive evaluation.**

**Major Comments:**

1. In the section 2.3.2, I believe authors should provide more details about the calculation process and algorithm implemented for the Sobol indices. And I believe it is too obvious that the sample sizes were dramatically different for different problems since they were calculating different variances based on different models, it is unnecessary to list these different size numbers.

More details about the Sobol indices calculation process were in the Appendix (A3). Following the comment above, those details were moved to the body of the text (line 365-384 in the revised manuscript), which now reads as:

"Parameter spaces were sampled using the Sobol quasi-random, cross-sampling strategy (Sobol, 2001). Rather than generating random numbers, this technique generates a uniform distribution in the probability space. The distribution appears qualitatively random, but sampling only takes place in regions of the probability function that were not previously sampled.

In order to assess the accuracy of the Sobol indices, confidence intervals of the indices should be constructed. The analytical procedure for confidence interval calculation involves repeating the model runs several times, which is too time consuming and computationally demanding in this case. Therefore the bootstrapping approach was used instead (Efron & Tibshirani, 1986). Archer et al. (1997) suggested using bootstrap confidence intervals to produce confidence intervals of complicated data structures. The bootstrapping approach is

based on resampling the parameter space of the already available data many times with replacement (randomly selecting values and allowing for duplicates), and constructing a distribution of the output (Efron & Tibshirani, 1986). Here, resamples were taken from the existing dataset with replacement, and the indices' values were recalculated. That gives an estimate of the mean and variance of each of the indices and allows calculation of the confidence interval. The method thus relies on computational cost rather than on an analytical cost (running the model again). Here, the samples used for the model evaluation were resampled 1000 times with replacement, and 95% confidence intervals were constructed (Archer et al., 1997).

Still, confidence intervals for the first-order indices (S1), with the Sobol sampling method gave values of more than 100%. This was also observed by Brunetti et al. (2016) and Hartmann et al. (2018) who also studied transport in unsaturated media. This may be a result of insufficient sample size, since Sobol's convergence requires a very large sample size (Saltelli et al., 2004). Therefore, here the S1 values were extracted using the delta method of Plischke et al. (2013), calculating S1 values from a given data through emulators and bootstrapping rather than running the model itself multiple times"

The details of the Sobol sample size were moved to the appendix.

2. I am not sure why the authors focused on the crashed simulations; it seems to be the problem of numerical model instead of sensitivity analysis problem. And the differences of GSA results using different methods to fulfill these "bad" input samples are more like pure numerical fluctuations to me.

The problem of crashed simulations is indeed a problem of numerical instability rather than a sensitivity analysis problem. However, as was demonstrated previously by Sheikholeslami et al. (2019), and by Clark and Kavetski, (2010) and as we also show in this study, these numerical artefacts are very difficult to tackle when running large number of simulations on large parameter space and they can affect the assessment of parameter sensitivities. This is because GSA sampling order and size is important for the GSA results. Therefore, it is important to devise solutions that minimize the effect of crashes on GSA (Razavi et al., 2021). Yet, very few strategies for handling simulation crashes have been proposed in the literature and identified for their shortcomings. This is why this data should be handled carefully and the way to treat this data in GSA is of importance.

3. In the section 3.2, have the authors tried the geostatistics tools to build different samples of heterogeneous media structure? I don't understand how the different samples representing the vadose zone media were generated. For the sentence in line 464, I believe Dai et al., (2017) has done some similar work and please check if it is helpful to improve this research for the heterogeneous media structure generation through geostatistics implementation.

We thank the reviewer for pointing our attention to Dai et al. (2017). Dai et al. (2017), used geostatistical approach to estimate the spatial distributions of three main parameters from point measurements. It was used to estimate the elevations of the contact between aquifer and aquitard from geological logs, the hourly head boundary conditions from monitored water elevations, and the hydraulic conductivity field from permeability data.

Dai et al. (2017), relied on the results of comprehensive field characterization measurements, and numerous field experiments and modeling studies performed at their site which provided them with a strong technical basis for their test case. They used that data to find the sources of parameters uncertainty through time and space in a 3-D model representing their site and solute transport in a ~200-hours tracers test.

Unlike the study of Dai et al. (2017), who examined the uncertainty of 3 parameters over a 3-D space in a specific site and limited time frame, in this study we deal with a more general problem examining the risk for contamination of the entire Israel's coastal plain vadoze zone over the course of 50 years and the importance of 17 parameters. While it is possible that geostatistcal tools could have provided a more realistic representation of the layers in each site, interpolation and representation of the rest of the space between the sites would require more data that we lacked and would have made the model and GSA much more computationally demanding than it already was. The more general objective of the study, has led us to adopt a more general 1-D representation of our space, where the ranges of the number of clay layers and their thickness were taken from field data.

For that purpose, we used a dataset of soil data of contaminated fuel sites along Israel coastal plain. From the dataset we extracted the average number and thickness of clay layers interbeds along the vadoze zone at each site, and calculated the total average for all sites. We then examined how much the number and thickness of these interbeds can affect benzene transport using the GSA.

Regarding line 464, we rephrased it to clarify we are dealing with clay interbeds and added the spatial distribution uncertainty tested by Dai et al., (2017) to the text (line 298-299 in the revised manuscript – moved to the M&M to the request of the 2nd reviewer) which now reads as follow:

"We are not aware of other studies that tested the distribution of clay layers interbeds in a SA for contaminants transport. Yet, Dai et al., (2017) tested the spatial distribution uncertainty of other parameters in a GSA like the elevations of the contact between aquifer and aquitard, the hourly head boundary conditions and the hydraulic conductivity field."

**Minor Comments:**

1. Line 84-85: I don't think these reference papers are all focus on the sensitivity analysis for the unsaturated zone.

   True, thank you very much for this remark - Ciriello et al., 2013 considered only saturated flow and was therefore removed. All other SA were done for unsaturated soils (line 69-70 of the revised manuscript).

2. Line 90: I don't understand the term "properties of benzene itself", what properties?

   Properties related to benzene and not only to the porous media like - the degradation rate or adsorption coefficient. Rephrased in the text as follow: "Yet, properties related to benzene itself (such as its degradation rate or adsorption coefficient)" (line 136 in the revised manuscript).

3. Line 99-100: I believe authors should provide more details about these two methods, especially about their algorithms, in the introduction.

We appreciate the comment. However, the other reviewer had asked us to shorten this paragraph on the GSA methods tested. Therefore we prefer to leave it this way with a comment that more details can be found in the method section – so as to have it somewhere in between the two requests of the two reviewers. All the details regarding the two methods appear in section 2.3 in the Materials and methods section. We added to the introduction a statement that more details regarding the two methods can be found in section 2.3 (line 152-155 in the revised manuscript) and it now reads as follow:

"Two GSA methods were tested for the homogenous media simulations, to analyse the parameter importance: the Morris method (Morris, 1991), a reliable, computationally-cheap alternative to variance-based GSA; and the Sobol method (Sobol, 2001), a computationally heavy, variance-based and probably the most well-established and widely-applied GSA (Khorashadi Zadeh et al., 2017; Jaxa-Rozen et al., 2021; Saltelli et al., 2010; Sobol, 2001; Song et al., 2015; Saltelli et al., 2004; Nossent et al., 2011; Brunetti et al., 2016, 2017, and many more) (see Material and Methods section 2.3 for a description of these methods)."

4. Line 103-104: please provide some references, I don't know this is a common problem.

(Razavi et al., 2021; Sheikholeslami et al., 2019) – were added as references (line 160 in the corrected manuscript). These authors suggested this is a common, though usually overlooked problem.

5. Some formats of titles of subsections are incorrect or inconsistent, please check all of them.

Thank you very much, we went over it and fixed it.

**Reviewer #2 comments on the manuscript "Identification of Parameter Importance for Benzene Transport in the Unsaturated Zone Using Global Sensitivity Analysis" submitted to *Hydrology and Earth System Sciences* (all authors answers are in blue font).**

I have reviewed the manuscript entitled "*Identification of Parameter Importance for Benzene Transport in the Unsaturated Zone Using Global Sensitivity Analysis,*" where the authors analyzed the sensitivity of Benzene transport to various physical and chemical parameters in both homogeneous and heterogeneous vadose zones. The analysis employed various sensitivity methods and addressed issues such as model crashes that can result in missing data and misinterpretation of sensitivity analysis outputs. This manuscript emphasizes a holistic approach to identifying sensitive parameters, which can guide future experimental design by focusing on the identified sensitive parameters. Therefore, it is of great interest for the HESS readers. However, there are some constructive and structural issues throughout the manuscript that must be resolved before publication. My comments and recommendations are below.

**We would like to thank the reviewer for the positive evaluation.**

**1. Line 50: Please provide a more general description of groundwater fuel contamination (not just in Israel).**

Some more information on groundwater fuel contamination can be found in the previous paragraph:

"Petroleum products such as gasoline and diesel are of the most abundant chemicals of ecological concern used nowadays. During petroleum exploration, production, transport and storage, petroleum products often find their way to the environment by accidental leaks and spills (Logeshwaran et al., 2018; López et al., 2008; Nadim et al., 2000). Consequently, groundwater is often polluted from surface sources, posing a substantial potential risk to potable water worldwide (López et al., 2008; Nadim et al., 2000; Logeshwaran et al., 2018; Reshef and Gal, 2017; Kessler, 2022). Since petroleum substances in general, and fuel components in particular, are considered toxic, carcinogenic, and mutagenic (Logeshwaran et al., 2018), strict regulations limit their maximum allowed concentration in groundwater to the parts per billion level (U.S EPA, 2006)."

 Following the reviewer's comment we also expanded this paragraph (line 50) and added more information related to the general context of fuel contamination **(added parts in bold font)**:

"Fuel products are usually comprised of different types of hydrocarbons. **Fuel compounds like benzene are among the most commonly found groundwater pollutants (Schmidt et al., 2004; Logeshwaran et al., 2018).** Benzene specifically, is highly soluble and thus of the most mobile fuel constituents in the subsurface (Farhadian et al., 2008). **In the U.S. alone, during 1987–1993 about 0.9 million kg were reported to be released into the terrestrial and aquatic environment by the petroleum industries (Fan et al., 2014).** In Israel, for example, benzene was detected in 60% of all sites monitored for fuel contamination (Reshef and Gal, 2017)…"

2. **Lines 59-60: Please add references and discuss what type of models were used. Elaborate on what parameters were used and how natural heterogeneity affects them.**

The following paragraph (lines 59-63 of the original text): "Most models simulating the transport of fuel contaminants in the unsaturated zone use a mechanistic description of the physical, chemical, and biological processes controlling contaminants' transport and attenuation. These models include many uncertain input parameters, due to the typical heterogeneity of the subsurface environment and the difficulty in obtaining sufficient relevant physical and bio-geochemical characterization of the site (Tartakovsky, 2007)" was changed to include and discuss comments 2 requirements and the studies requested in comment 5 below (lines 62-81 of the revised manuscript):

Since actual water flow and contaminants transport in the subsurface are difficult to measure and predict, mathematical models are used to solve such transport problems (Bear and Cheng, 2010). Many studies have been performed to examine the fate and transport of petroleum hydrocarbons, and specifically of benzene, most in saturated homogeneous porous media (Lu et al., 1999; Brauner and Widdowson, 2001; Choi et al., 2005, 2009). Few studies have also dealt with the movement of benzene in unsaturated porous media (Berlin et al., 2016; Berlin and Suresh, 2019; Troldborg et al., 2009; Ciriello et al., 2017). Benzene transport models typically combine Darcy-type water flow, advective–dispersive transport and a source/sink term considering various physical, chemical and biological processes including sorption, dissolution, and biodegradation (Mohamed and Sherif, 2010). Yet, due to the typical heterogeneity of the subsurface environment and the difficulty in obtaining sufficient relevant physical and bio-geochemical characterization of the site, these parameters entail high level of uncertainty (Ciriello et al., 2017; Tartakovsky, 2007). Soil heterogeneity and layering, for example, was shown to have considerable effect on contaminants transport in some studies (Rivett et al., 2011; Chen et al., 2019), while in others the effect seems to be negligible (Botros et al., 2012; Akbariyeh et al., 2018). This of course depends on the model scale, type of system (natural or irrigated), and type of contaminant tested. Also, the estimation of soil and contaminants parameters can be done in various methods, such as laboratory measurements, scaling and inverse modelling (Botros et al., 2012; Akbariyeh et al., 2018; Berlin et al., 2016; Berlin and Suresh, 2019; Troldborg et al., 2009; Ciriello et al., 2017). This adds another aspect of uncertainty to the model.

3. **Lines 67-68: Please add References. Many disciplines are using these methods. Do they all use it the same? Are they all reporting similar challenges as in the current study?**

References Saltelli and Annoni, 2010; Razavi et al., 2021 were added (lines 86-87 of the revised manuscript).

Regarding the use of GSA in different disciplines, we believe a detailed discussion of the subject is beyond the scope of this specific research and of the introduction section. However, we have added text regarding the main challenges in the application of GSA in hydrological modelling as identified by Song et al. (2015) (lines 120-128 of the revised text):

"In his review, Song et al. (2015) identifies three main "hot spots" in GSA application with hydrological modelling, though these are relevant to many other disciplines. The three "hot spots" are: (1) Computational cost and subsequent meta-modelling used instead of running

the models multiple times, where the reliability and goodness-of-fit of meta-models should be explored, (2) GSA method selection, convergence and reliability - selecting an appropriate GSA method, monitoring the convergence, and estimating the uncertainty of the GSA results are important for hydrological models, (3) GSA methods involve many hypotheses or have other limitations, including the independence of input variables, where in practice, the parameters employed by hydrological models usually have interactions or correlations that needs to be considered."

**4. Line 86: replace the 'papers' by 'studies'**

Done, thank you (line 130 of the revised manuscript).

**5. Line 94: There are studies on transport in heterogeneous media that should be discussed.**

1. **Akbariyeh, S., Bartelt-Hunt, S., Snow, D., Li, X., Tang, Z., & Li, Y. (2018). Three-dimensional modeling of nitrate-N transport in vadose zone: Roles of soil heterogeneity and groundwater flux. Journal of contaminant hydrology, 211, 15-25.**
2. **Botros, F. E., Onsoy, Y. S., Ginn, T. R., & Harter, T. (2012). Richards equation–based modeling to estimate flow and nitrate transport in a deep alluvial vadose zone. Vadose Zone Journal, 11(4).**
3. **Chen, N., Valdes, D., Marlin, C., Blanchoud, H., Guerin, R., Rouelle, M., & Ribstein, P. (2019). Water, nitrate and atrazine transfer through the unsaturated zone of the Chalk aquifer in northern France. Science of the Total Environment, 652, 927-938.**

Added, see comment 2 above.

**6. Lines 95-96: What does it mean '*individual impact of multiple parameters*'?**

It means how much each parameter of the set of parameters tested affects the model. For clarity purposes the sentence was rephrased to: "The objective of this study was to assess the specific impact of each of the multiple parameters that affect benzene transport in the unsaturated zone" (line 140 of the revised manuscript).

**7. Lines 96-108: The objectives should be concise. You can elaborate on the choice of a specific method in the method section.**

It is true. Yet, the other reviewer had asked us to elaborate more in this paragraph on the GSA methods tested. Therefore we prefer to leave it this way with a comment that more details can be found in the method section – so as to have it somewhere in between the two requests of the two reviewers (line 155 of the revised manuscript).

**8. Lines 167-171: 1. Is there a chemical\biological reason?  2. I didn't understand the range issue (why not analyze using a logarithmic scale?)**

1. No chemical or biological issue here, merely an issue of sensitivity. As stated in the text, high values of $\lambda_k$ always resulted in concentration and flux of 0, which lowered the overall sensitivity and analytical strength of the GSA.

2. The Morris method always takes the range tested and divides it in a very simple manner to several equal parts, it does not use a logarithmic scale.

9. **Lines 180-184: You should plot the range of profiles and the map, if there is one, of all the locations. Put in supporting information.**

Unfortunately we cannot plot the range since only a general range of 2-50 m groundwater depth was stated in the report for the 236 contaminated sites tested (Reshef and Gal, 2017). The below map is given in the report. It shows the distribution of monitored fuel facilities and the state of contamination on site. The map was added to the supporting information section (Figure S1). Now mentioned in line 248 of the revised manuscript.

[Figure]

Distribution of monitored fuel facilities and the state of groundwater contamination in circles (red – fuel plume, black – severe contamination, blue – clean, green – minor contamination)

10. **Line 200: There shouldn't be a single-line paragraph.**

Thank you. This paragraph was combined with the paragraph below (line 267-268 of the revised manuscript).

**11. Line 204: Only clay layers. How do you define Clay? by texture percentage?**

Following comment 15 below, the details regarding clay layers definition and distribution were moved from results to the methods part section 2.2 where the answers to the above mentioned question are given (line 275 of the revised manuscript).

**12. Lines 211-212: Delete**

Deleted.

**13. Lines 332-336: Delete**

Deleted.

**14. Lines 339-346: Move to 'Methods'**

We believe this part is important and appears at the beginning of the results for each GSA preformed, since it gives general information on the GSA itself and how much of the runs were converged. This has important implications on the results.

**15. Lines 442-464: Move to 'Methods'**

This section was moved to the methods part - line 275 of the revised manuscript.

**16. Line 482: replace 'adress' by 'address'**

Replaced – thank you very much (line 623 of the revised manuscript).

**References**

[revised manuscript text omitted]

---

## Author Response (AR2)

**Response to editor and reviewers comments for the manuscript "Identification of Parameter Importance for Benzene Transport in the Unsaturated Zone Using Global Sensitivity Analysis" submitted to *Hydrology and Earth System Sciences* (all authors' answers are in blue font, all editor's and reviewers' comments are in black font).**

**Editor comments:** While I do agree with the Reviewer, I would suggest the Authors to carefully check their manuscript also with reference to proper usage of English and in light of the very final comments provided by the Reviewer. I will then be in a position to make final assessment.

We agree with the editor's comment. The manuscript was sent for final English and scientific revision these changes can be found in the manuscript track changes version across the text. Also a rerun of the heterogeneous media Morris analysis was conducted as detailed below.

Please find the comments given by the reviewer and our response below.

**Reviewer comments:**

Dear Editor
I have once again reviewed the manuscript titled 'Identification of Parameter Importance for Benzene Transport in the Unsaturated Zone Using Global Sensitivity Analysis.' The authors have addressed the majority of the comments, and I believe the manuscript is now ready for publication.

We thank the reviewer for his positive evaluation.

However, I did notice a few mistakes and typographical errors, as outlined below. Moreover, I do suggest that the authors add a short discussion (no more than four lines) on why they think the λ is the most sensitive and under what conditions it would be so sensitive. I think it would draw more attention.

The following text was added: "$\lambda_k$ the rate of benzene removal from the media by biological degradation. This rate can vary greatly from extremely fast to very slow rates, depending on parameters such as initial benzene concentration and soil water content (Table S1). In general, the degradation rate is lower for higher concentrations and lower water content, and vice versa." (lines 621-624 of the revised manuscript).

Following the reviewer's comment the manuscript was sent to professional English editing. We believe the manuscript is now ready for publication.
Specific comments:

1. Line 182: The q is separately calculated using the Darcy-Buckingham? The q is calculated by solving the Richards equation. I understand that the Darcy-Buckingham is a component within the Richards equation, but technically you don't solve the Darcy-Buckingham directly.

We agree with the reviewer. q is calculated by solving the Richard's Equation. The definition of q is given here for completeness.

$$\theta D^w = \alpha_L q + \theta D^M \tau_w \qquad (1)$$

where $D^M$ is the benzene's molecular diffusion coefficient in the aqueous phase [L$^2$T$^{-1}$], $q$ is the absolute value of the Darcian fluid flux [LT$^{-1}$] evaluated using the Darcy–Buckingham law, $q = -K(\frac{dh}{dz} + 1)$. "

Table 2: the range of the n parameter for clayey soil is wrong (0.13-1.31). We know that n cannot be smaller than 1.

It is true and we would like to thank the reviewer for this important comment. Following this comment the lower value of $n_2$ for the clayey soil was updated in Table 2 to a range of 1.09-1.31 (Carsel and Parrish, 1988). Also we reran the Morris sensitivity analysis for heterogeneous media with the updated values. The parameters ranking came up a bit different, with higher weight of some of the clay soil parameters, smaller error bars, more cases of the model incovergence and results that were overall more coherent with the homogenous media results and the literature.

The corrected results are presented in section 3.2 and are also addressed in the abstract (lines 34-38 and lines 40-42 of the revised manuscript) and in the summary and conclusions (lines 655-658 of the revised manuscript).

2. Line 491: revise the capital T to small t.

Corrected, thank you (line 499 of the revised manuscript).

**References**

Carsel, R. F. and Parrish, R. S.: Developing joint probability distributions of soil water retention characteristics, Water Resour. Res., 24, 755–769, https://doi.org/10.1029/88WR01772., 1988.